# Drivers of uncertainty in future projections of MJO teleconnections

Andrea M. Jenney[1,2], David A. Randall[2], and Elizabeth A. Barnes[2]

[1]Department of Earth System Science, University of California, Irvine, CA, USA
[2]Department of Atmospheric Science, Colorado State University, Fort Collins, CO, USA

**Correspondence:** Andrea M. Jenney (ajenney@uci.edu)

**Abstract.** Teleconnections from the Madden-Julian Oscillation (MJO) are a key source of predictability of weather on the extended time scale of about 10-40 days. The MJO teleconnection is sensitive to a number of factors, including the mean dry static stability, the mean flow, and the propagation and intensity characteristics of the MJO, which are traditionally difficult to separate across models. Each of these factors may evolve in response to increasing greenhouse gas emissions, which will impact MJO teleconnections and potentially impact predictability on extended time scales. Current state-of-the-art climate models do not agree on how MJO teleconnections over central and eastern North America will change in a future climate. Here, we use results from the Coupled Model Intercomparison Project Phase 6 (CMIP6) historical and SSP585 experiments in concert with a linear baroclinic model (LBM) to separate and investigate alternate mechanisms explaining why and how boreal winter (January) MJO teleconnections over the North Pacific and North America may change in a future climate, and to identify key sources of inter-model uncertainty. LBM simulations suggest that a weakening teleconnection due to increases in tropical dry static stability alone are robust across CMIP6 models, and that uncertainty in mean state winds is a key driver of uncertainty in future MJO teleconnections. Uncertainty in future changes to the MJO's intensity, eastward propagation speed, zonal wavenumber, and eastward propagation extent are other important sources of uncertainty in future MJO teleconnections. We find no systematic relationship between future changes in the Rossby wave source and the MJO teleconnection, or between changes to the zonal wind or stationary Rossby wave number and the MJO teleconnection over the North Pacific and North America. LBM simulations suggest a reduction of the boreal winter MJO teleconnection over the North Pacific, and an uncertain change over North America, with large spread over both regions that lends to weak confidence in the overall outlook. While quantitatively determining the relative importance of MJO versus mean state uncertainties in determining future teleconnections remains a challenge, the LBM simulations suggest that uncertainty in the mean state winds is a larger contributor to the uncertainty in future projections of the MJO teleconnection than the MJO.

## 1 Introduction

As the most energetic mode of tropical intraseasonal variability, the Madden-Julian Oscillation (MJO) is one of the most important sources of global weather predictability on the extended time range of about 10-40 days (Robertson et al., 2015).

While previous work suggests that models agree on how MJO teleconnections will change in a future climate over specific regions (e.g., many models predict a strengthening of the MJO teleconnection over the North American west coast), over much of the North Pacific and over North America, it is unclear how the influence of the MJO will evolve with increasing atmospheric greenhouse gases (Zhou et al., 2020).

    Whereas some consensus has been emerging about at least one of these controls that affects global MJO teleconnection

strength–the increase of the tropical dry static stability with surface warming–recent studies suggest this control may be secondary (Bui and Maloney, 2018; Maloney et al., 2019; Bui and Maloney, 2019a, b; Zhou et al., 2020). To first order, the rate that air rises and sinks in the MJO's circulation is tightly constrained by, and inversely proportional to, the tropical dry static stability (Wolding et al., 2016). Latent heat release associated with MJO precipitation is balanced by the upward advection of dry static energy. Similarly, radiative cooling in the MJO's dry region is balanced by slow, adiabatic subsidence. That is,

the MJO's circulation strength is directly proportional to its diabatic heating rate, and inversely proportional to the dry static stability. It is expected that the tropical static stability will increase in the future as the tropical temperature profile adjusts towards the moist adiabatic lapse rate of a warmer surface (e.g., Santer et al., 2005). Ignoring changes to the MJO's precipitation intensity or its cloud optical properties (i.e., its diabatic heating rate), this will weaken the MJO's vertical circulation and its associated upper-level divergence. In simulation studies of a future climate forced with increasing greenhouse gases,

many models predict a weakening of the MJO's circulation strength (Bui and Maloney, 2018; Maloney et al., 2019). However, there is a considerable amount of inter-model spread, much of which is tied to disagreement in future projections of the MJO's diabatic heating and precipitation (Bui and Maloney, 2018; Maloney et al., 2019; Bui and Maloney, 2019a, b). While previous work has attributed a simulated weakening of MJO teleconnections in one model to the increase in dry static stability (Wolding et al., 2017), a recent model intercomparison study has shown that in climate change simulations the MJO's circulation change

is not correlated with its teleconnection changes (Zhou et al., 2020).

    Samarasinghe et al. (2021) suggest that the lack of an apparent relationship between modeled MJO circulation change and teleconnection change may be related to changes in the mean state winds. Mean state winds refer to the winds that vary slowly in time (i.e., on seasonal, rather than daily time scales). Henderson et al. (2017) emphasize the importance of the mean state winds in the MJO teleconnection pattern, showing that biases in modeled MJO teleconnections can be attributed, in part, to

biases in the mean state winds. Mean state winds affect both Rossby wave excitation and their propagation. The magnitude of the vorticity and its horizontal gradient on the equatorward flank of the jet play an important role in the excitation of Rossby waves by the MJO (Sardeshmukh and Hoskins, 1988; Mori and Watanabe, 2008; Seo and Son, 2012; Seo and Lee, 2017; Zheng and Chang, 2020). The jets also act as a waveguide, and their structure and intensity determine the path that Rossby waves take as they propagate, and the sizes of Rossby waves that can propagate through them (e.g., Hoskins and Ambrizzi, 1993;

Karoly, 1983). Tseng et al. (2020b) and Zheng and Chang (2020) show how variability in the mean state winds over the eastern North Pacific drives variability in the MJO teleconnection, with variations in the jet's zonal extension or meridional position playing an important role in modulating wave propagation, and hence MJO teleconnectivity, over North America. Finally, Zhou et al. (2020) show that modeled increases in the MJO's impact over the North American west coast with warming are due to an extension of the Pacific jet, which causes a shift in the MJO teleconnection pattern. However, little work has been done to

explore the direct role that changes to the mean state winds have on changing MJO teleconnections to the broader North Pacific and North America region.

In addition to features of the mean state (its static stability and winds), the MJO teleconnection may also respond to changes in the MJO. The MJO teleconnection is known to be sensitive to its eastward propagation speed. While one study found little sensitivity of extratropical teleconnections to the eastward propagation speed of tropical heating (Goss and Feldstein, 2017), many studies have found the opposite. Comparisons between composites of observed geopotential height anomalies during fast and slow MJO events show differences in both the teleconnection pattern and magnitude of the anomalies (Yadav and Straus, 2017). Models with faster propagating MJOs have weaker MJO teleconnections (Wang et al., 2020; Zheng and Chang, 2019). In a simulation study, Bladé and Hartmann (1995) showed that an eastward propagating heat source is associated with a weaker and smaller wavetrain than a stationary or westward propagating heat source. To first order, this is a linear effect: eastward (westward) propagation effectively embeds the heating in easterlies (westerlies) (see Sect. 2b of Bladé and Hartmann, 1995). Many climate models predict an increase in the MJO's propagation speed with warming (e.g., Rushley et al., 2019), which, considering previous work, may contribute to a weakening, or at least a change in the pattern of the MJO teleconnection with warming.

In the present climate, MJO convection is generally confined to the Indian and West Pacific oceans, while the MJO circulation signal circumnavigates the tropics. In simulations of a future warmer climate, many models predict an increase in the eastward extent of MJO convection (Adames et al., 2017; Bui and Maloney, 2018; Chang et al., 2015; Subramanian et al., 2014). This may be driven by the expansion of the eastward edge of the Indo-Pacific warm pool, a region of very warm tropical sea surface temperatures (Maloney and Xie, 2013; Zhou et al., 2020). A more eastward propagating MJO may extend the region of Rossby wave excitation by the MJO further eastward as well.

Beyond MJO's precipitation intensity, propagation speed, and propagation extent, the MJO's zonal wavenumber is projected to decrease (Rushley et al., 2019). Previous work has also found an increase in the frequency of MJO events (e.g., Arnold et al., 2013; Chang et al., 2015; Cui and Li, 2019). Lastly, the meridional extent of the MJO may also change, although it is unclear if this is expected with surface warming.

Studies exploring the MJO teleconnection change with warming in global climate models are sparse. Samarasinghe et al. (2021) find strengthened future MJO teleconnections over the Gulf of Alaska and North Atlantic in a climate model forced with high emissions. While Zhou et al. (2020) show that a multi-model mean change in the MJO's impact on both the circulation and precipitation over most of North America is near-zero, they do not quantify the uncertainty in this projection over this larger region. Their results for the North American west coast, which rely on a subset of the publicly available climate model data (they analyze only those models that produce the most realistic MJOs), hint that while the multi-model mean change over broader North America is small, there may be quite a bit of disagreement between models.

A detailed study of potential MJO teleconnection changes is attractive. We are motivated to unpack how and why MJO teleconnections might change under global warming, and here seek to clearly and quantitatively rank the various controls on MJO teleconnection strength changes in a warmer climate to better understand how the MJO teleconnection might change in

the future. Additionally, we desire to maximize the resources offered by the latest phase of the Coupled Model Intercomparison

Project (CMIP6), by avoiding having to subset only those models producing more realistic MJOs.

In this study, we will use a linear baroclinic model (LBM, Watanabe and Kimoto, 2000) in concert with output from CMIP6 to separately quantify the contributions to uncertainty in future projections of the MJO teleconnection by various mechanisms: mean state changes and changes to the MJO's eastward propagation extent, propagation speed, heating intensity, and zonal wavenumber. The use of the LBM for studying the MJO teleconnection has precedent. Previous work shows the MJO telecon-

nection is, to first order, linear (Mori and Watanabe, 2008; Lin and Brunet, 2018). Many previous studies have used the LBM to untangle MJO teleconnection mechanisms (e.g., Henderson et al., 2017; Tseng et al., 2019, 2020a, b; Wang et al., 2020; Wolding et al., 2017). The linear model additionally has a few features that make it particularly attractive for the current study. First, it is very inexpensive to run, allowing us to cheaply and efficiently run hundreds of perturbation simulations, a task that is difficult and expensive using complex global climate models. Second, the linear framework is quite simple in that almost

all simulated variability is from an external forcing, which makes the interpretation of the results very straightforward. Third, because the mean state is maintained by prescribing it at each time step, it is possible to run simulations with the temperature field out of balance with the wind field. This point allows us to separately and cleanly quantify the uncertainty in the MJO teleconnection change from uncertainty in the thermal structure of the atmosphere versus its mean winds. Finally, because we can separately prescribe an MJO forcing, it is not necessary to have a realistic internally generated MJO when investigating the

sensitivity of the MJO teleconnection to inter-model variations in the mean state, thus permitting us to avoid only analyzing those CMIP6 models which produce more reliable MJOs.

## 2    Methods

We explore the sensitivity of the MJO teleconnection to changes in the MJO's propagation speed, eastward propagation extent, zonal wavenumber, heating magnitude, and of the mean state dry static stability and winds using simulations with the LBM

of Watanabe and Kimoto (2000). The time-integration version of the LBM that we use here is a spectral model that solves linearized equations for vorticity, divergence, temperature, and surface pressure. The model takes two inputs: a mean state and a forcing (which is added to the simulated perturbation field at each time step).

The model equations are described in appendix B of Watanabe and Kimoto (2000). We use a horizontal truncation of T42 (roughly 2.8° horizontal resolution) and 20 vertical levels. The model is numerically damped with biharmonic horizontal

diffusion, for which we use an e-folding time of 10 minutes for the smallest resolvable waves, and with linear drag, with a damping time scale of 0.5 days applied to the lowest 3 model levels, 1 day for the top two model levels, and 20 days for all levels in between. We use this strong damping to inhibit the growth of baroclinic waves, particularly because in some simulations we intentionally prescribe mean state winds that are not in balance with the mean state temperature. As a consequence of the strong damping and linear framework, eddy convergences of heat and momentum in the extratropical storm tracks, including

those associated with MJO variability (e.g., Deng and Jiang, 2011; Guo et al., 2017; Takahashi and Shirooka, 2014), are not

explicitly accounted for. However, because the mean state wind used in the LBM simulations is taken from CMIP6, eddy-mean flow interactions are, in a way, included implicitly.

We will make the convention for composite analysis that "historical" and "future" mean states refer to January conditions during 1984-2014, and 2071-2100, respectively, of the historical and SSP585 simulations from models participating in CMIP6. SSP585 is a relatively high atmospheric greenhouse gas emissions scenario following shared socioeconomic pathway 5 (SSP5; O'Neill et al., 2017) and representative concentration pathway 8.5 (Moss et al., 2010).

We restrict our analysis to those 29 CMIP6 models for which there was monthly mean data for the variables required as input by the LBM on the data node at https://esgf-node.llnl.gov/search/cmip6/ at the time of access. We use the r1i1p1f1 member for all models except for GISS-E2-1-G (r1i1p1f2), HADGEM3-GC31-LL (r1i1p1f3), UKESM1-0-LL (r1i1p1f2), and MIROC-ES2L (r1i1p1f2). We acknowledge that sampling ensemble spread is desirable in order to isolate the forced response from internal variability. Thus, for each CMIP6 model mean state combination, we construct an ensemble of 30 separate LBM simulations, each using a different January. We use interannual ensembles in this way, rather than using the other ensemble members from each CMIP6 model, to minimize the data downloading burden and to keep the number of ensemble members used for each CMIP6 model mean state consistent. Lastly, the SSP585 simulation of CAMS-CSM1-0 was only carried out until 2099, and so for this model we use only 29 years.

We force the LBM with a propagating idealized horizontal dipole heating, which is preferable to a realistic heat source (for example, one obtained from composites of observed MJO events), since it allows easy manipulation of the MJO's intensity and propagation characteristics. The idealized heating (as in Ting and Held, 1990; Seo and Son, 2012) is designed to be MJO-like, with the perturbation temperature given by

$$T'(\lambda, \phi, z, t) = A e^{\frac{-\sin^2 \phi}{2\sigma^2}} \sin(k\lambda - \omega t) e^{\frac{-(\lambda - b)^2}{2c^2}} \sin^2 \left( \frac{\pi(z - z_{top})}{z_{bot} - z_{top}} \right), \tag{1}$$

where $T'$ is the perturbation temperature of the forcing, $\phi$ and $\lambda$ are the latitude and longitude, respectively, $z$ is the model vertical coordinate with subscripts $bot$ and $top$ referring to the lowest and highest model level, respectively, $A$ is the heating amplitude, $k$ is the zonal wavenumber, $\omega$ is the temporal frequency of the heating, and $t$ is the time. Unless otherwise noted, we set $A = 0.1$ K day$^{-1}$, $\sigma = 0.1$, $k = 1.9$, $\omega$ corresponding to a temporal period of 40 days, $b = 130°$, and $c = 45°$. Figure 1 shows the perturbation heating ($T'$) in the mid-troposphere where it is a maximum for $t = 10$ days, 20 days, 30 days, and 40 days. To impose a realistic MJO horizontal scale, and facilitate varying it, we constrain the propagating heat source to the longitudes over the Indian and West Pacific Oceans by multiplying the sinusoidal heat source by a Gaussian centered over the Maritime continent (see Eq. 1). We initialize the model with the forcing given by setting $t = 1$ day, and integrate for 60 days.

The experiments are described in Table 1. When testing the sensitivity of the MJO teleconnection to perturbations of the mean state or of an MJO intensity or propagation characteristic, we perturb only the feature being tested and hold everything else constant. For each perturbed feature, we run two sets of experiments: one with the mean climate given by historical values, and the other with the mean climate given by future values (see the "sub-experiment" column in Table 1). For example, in testing the sensitivity of the teleconnection to the mean state winds, we run two sets of experiments: one set in which the dry

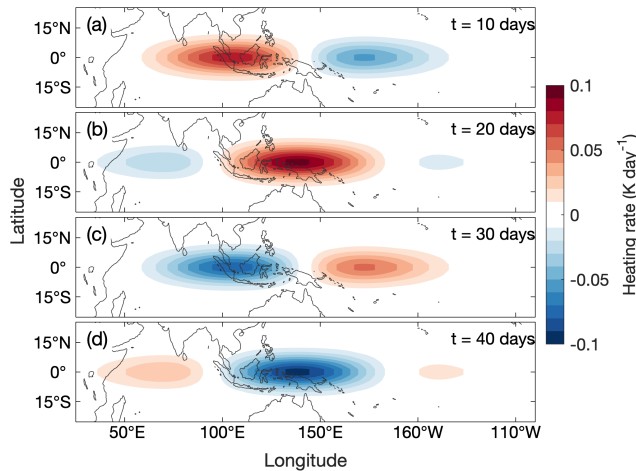

**Figure 1.** Propagating thermal forcing used in the LBM simulations at $t$ = (a) 10 days, (b) 20 days, (c) 30 days, and (d) 40 days.

static energy is held constant at historical values while the winds are perturbed, and the other set with the dry static energy held

constant at future values while the winds are again perturbed. The LBM takes as input three-dimensional winds, temperature, geopotential height, and surface pressure. The set of variables held constant when the "winds" are held constant are the zonal, meridional, and vertical wind. Similarly, the set of variables held constant when the "dry static energy" is held constant are the temperature, geopotential height, and surface pressure. The use of the many CMIP6 mean states permits the quantification of uncertainty in the future MJO teleconnection due to "model uncertainty" (e.g., Lehner et al., 2020) of the mean state.

We are guided by the results of previous studies for the experiments where the full mean state is held constant and MJO intensity or propagation characteristics are being perturbed. For these simulations, we aim to quantify uncertainty by obtaining the range of possible changes to the MJO teleconnection that result from changes to the MJO. We do this by conducting experiments with perturbations representing the lower and upper bound of changes to the MJO that are simulated by models participating in the Coupled Model Intercomparison Project Phase 5 (CMIP5) at the end of the century given high emissions.

For climate models that simulate realistic MJOs in CMIP5, Rushley et al. (2019) find an increase in MJO propagation speed between 1.8 and 4.5 % K$^{-1}$, and a maximum decrease of the zonal wavenumber of about 0.2 (we use no change in the zonal wavenumber as a lower bound). Using the increase of the multi-model mean, global mean surface temperature between the historical and future climates of the models used in this study, we thus conduct experiments where the propagation speed of our idealized forcing is perturbed to correspond to an MJO period of about 37.5 (34) days for the lower bound (upper bound)

experiment where sensitivity to MJO propagation speed is being tested. In modification of the MJO's eastward extent, we use no change as a lower bound and for the upper bound, extend by 20° the eastern edge of the gaussian used to confine the propagating forcing to the Indian and West Pacific Oceans. Specifically, the gaussian multiplier of the perturbation forcing is the control gaussian with the maximum value sustained for an additional 20° to the east before decreasing. This value is roughly informed from results in Maloney et al. (2019) (CMIP5) and in Zhou et al. (2020) (CMIP5 and CMIP6). Lastly, we are informed

by Bui and Maloney (2018), who find that the subset of CMIP5 models that produce MJOs validating best against present-day

observations tend to simulate changes of the MJO's precipitation amplitude between -10% to + 20% between the historical and future climates under RCP8.5 forcing. In the experiment testing sensitivity to MJO amplitude, we use perturbed MJO heating amplitudes of 0.9 and 0.12 K day$^{-1}$, corresponding to a 10% reduction and 20% amplification of the control forcing. We use estimates of changes to the MJO from CMIP5, rather than CMIP6, due to the availability of published, detailed analyses of MJO changes for CMIP5. However, we note that despite recent improvement in the simulation of the MJO in CMIP6 (Ahn et al., 2020) much work is needed to deepen understanding of future changes to the MJO, and thus these estimates are crude, first guesses. We include a figure showing the perturbed MJO cases in Appendix A.

With the described setup, some of the simulations became unstable. We readily admit this is an inescapable trade-off of having implemented a forced linear model framework; despite its advantages for rapidly sampling and intercomparing first-order effects, the neglect of secondary nonlinear buffering mechanisms can corrupt some integrations. Pragmatically, we thus omit all LBM simulations in which, at any location, the maximum value of geopotential height at 500 hPa during the second half of the integration is more than ten times the maximum value during the first half. Additionally, we omit all LBM simulations using a specific CMIP6 model's mean state if 10 or more of the 30 constituent ensemble simulations for any of the mean state combinations meets the instability criterion described above, leaving 24 of the 29 CMIP6 models for analysis. Figure A2 in Appendix A shows, for each basic state combination, the fraction of ensemble members that became unstable. The last two rows of Table 1 list the number of total simulations run for this study (minus those from the five CMIP6 models we omit) and the number of simulations we omit from our analysis for each experiment (which is reassuringly less than 5% of the total simulations for each experiment). Note that the "number of unique simulations" field is blank for the mean state winds feature (second row) because the simulations are shared with that of mean state dry static stability (first row).

Lastly, we emphasize that the linear framework we are using neglects nonlinear interactions in simulating MJO teleconnections. For example, in reality, the mean state exerts a strong control on MJO propagation characteristics (Jiang et al., 2020). Rossby waves excited by the MJO are able to extract energy from the mean flow (Adames and Wallace, 2014; Zheng and Chang, 2020), and nonlinear interactions have been shown to lead to spatial shifts in extratropical teleconnection patterns (Lin and Brunet, 2018). Thus, we caution that the results of this study are limited by the exclusion of these effects.

## 3  Results

### 3.1  Mean state

We conduct experiments with the experimental setup described by Table 1.

Reassuringly, the propagating MJO-like forcing in the model excites a plausible time-varying response in the extratropics. As an example, Fig. 2a shows the time-varying response of 500 hPa geopotential height at 60° N, 115° W for each ensemble member (thin light blue lines) and their mean (thick black line) for the set of 30 simulations performed with the full mean state given by historical values from a single representative CMIP6 model (ACCESS-ESM1-5). The smallness of the geopotential height anomalies relative to a typical observed midlatitude value is an expected consequence of the smallness of the amplitude used to force the model, which we kept small to try to minimize the number of unstable simulations.

**Table 1.** Experimental setup for the linear baroclinic model simulations. "Lower bound" and "upper bound" refer only to perturbations of the thermal forcing (last three rows). DSE = dry static energy.

| Feature | Sub-experiment | Control | Perturbation (lower bound) | (upper bound) | Number of unique simulations | Number of unstable simulations |
|---|---|---|---|---|---|---|
| Mean state DSE | Historical wind | Historical DSE | Future DSE | | 1440 | 49 |
| | Future wind | | | | 1440 | 63 |
| Mean state winds | Historical DSE | Historical wind | Future wind | | | |
| | Future DSE | | | | | |
| Forcing heating magnitude | Historical mean state | $A = 0.10$ K day$^{-1}$ | $A = 0.09$ K day$^{-1}$ | $A = 0.12$ K day$^{-1}$ | 60 | 0 |
| | Future mean state | | | | 60 | 0 |
| Forcing propagation speed | Historical mean state | $\omega = 1/40$ day$^{-1}$ | $\omega = 1/37.5$ day$^{-1}$ | $\omega = 1/34$ day$^{-1}$ | 600 | 0 |
| | Future mean state | | | | 600 | 0 |
| Forcing eastward extent | Historical mean state | | $0°$ extension | $20°$ extension | 600 | 0 |
| | Future mean state | | | | 600 | 1 |
| Forcing wavenumber | Historical mean state | $k = 1.9$ | $k = 1.9$ (no change) | $k = 1.7$ | 600 | 2 |
| | Future mean state | | | | 600 | 3 |

We next introduce a scalar metric of MJO teleconnection strength that is appropriate to visualize in map form, recognizing that the magnitude of peaks and troughs of the ensemble mean response to the propagating forcing (for example, the thick black line in Fig. 2a) is one way to quantify the magnitude of the consistent (ensemble mean signal that stands apart from the noise due to internal variability) teleconnection strength. Given that the amplitude of an infinite sine or cosine wave is the square root of twice its variance, we define the teleconnection strength at each point as the square root of twice the temporal variance of the ensemble mean geopotential height at 500 hPa, ignoring the first 10 days of each simulation to avoid sampling conditions prior to the establishment of representative spread between interannually varying ensemble members. That is, we take the ensemble mean first and then calculate the variance. We will refer to this value as the "amplitude" of the modeled MJO teleconnection. This metric is qualitatively similar to the STRIPES index of Jenney et al. (2019), which is useful for quantifying the sensitivity of the extratropics to MJO variability across all MJO phases.

The baseline multi-model MJO teleconnection simulated by the LBM looks reassuringly plausible. Figure 2b shows the multi-model mean MJO teleconnection amplitude for the set of LBM simulations performed with both the mean state dry static energy (the vertical gradient of which gives the dry static stability) and winds given by historical values. Darker colors indicate that MJO teleconnectivity is larger. The MJO-like forcing excites Rossby waves that propagate in high density through the Pacific waveguide, leading to large values of teleconnectivity there. The pattern of MJO teleconnectivity over North America, which exhibits maxima over the southeastern and northwestern region, with a minimum over the central region, is consistent with the pattern of the observed boreal winter MJO influence on North American geopotential height, temperature, and sea level pressure (e.g., Jenney et al., 2019; Zhou et al., 2012; Samarasinghe et al., 2021).

The remainder of Fig. 2 reveals, in general, a reduction of MJO teleconnectivity due to future mean state effects, decomposed into separate contributions from stability vs. winds. Figure 2c-e shows the multi-model difference in the amplitude of the modeled MJO teleconnection between the set of simulations with the historical mean state and the set of simulations with future perturbations to the mean state. The forcing used in all simulations is the same. Panel (c) shows the change to the multi-model mean teleconnection that results from future changes to the winds, alone. Over most of North America, the change in the teleconnection is near-zero. Over the eastern Pacific and Bering Sea, changes to the winds lead to a weaker teleconnection. Other studies have noted an increase in the MJO's boreal winter influence near California (Zhou et al., 2020), attributing this to an eastward extension of the Pacific jet. This strengthening is not apparent here. Panel (d) shows the change to the multi-model mean teleconnection that results from future increases in tropical dry static stability, alone. That is, changes to the winds are not taken into account and are held constant between these two sets of simulations. The teleconnection is weaker in the simulations with future dry static energy because the higher static stability weakens the MJO circulation and subsequent Rossby wave source (Sardeshmukh and Hoskins, 1988) that results from the prescribed thermal forcing.

Panel (e) of Fig. 2 shows the sensitivity of the MJO teleconnection to a change in the full mean state, neglecting any changes to the MJO. In isolation (apart from changes to the MJO), changes to the mean state weaken the multi-model mean teleconnection amplitude in places where the teleconnection is the strongest. This is mostly due to an increase in the tropical dry static stability (panel d), but over the Eastern Pacific is also due to to changes to the winds (panel c). Figure 3, which shows

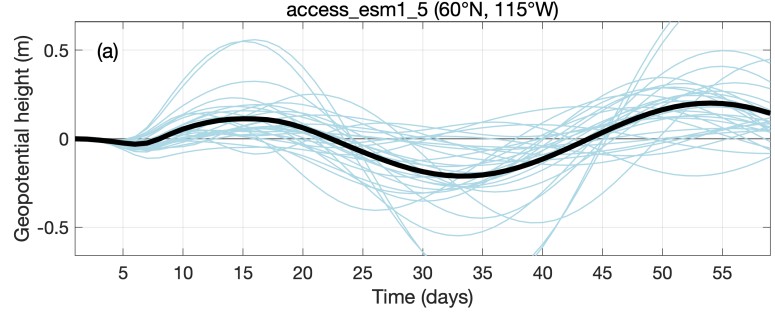

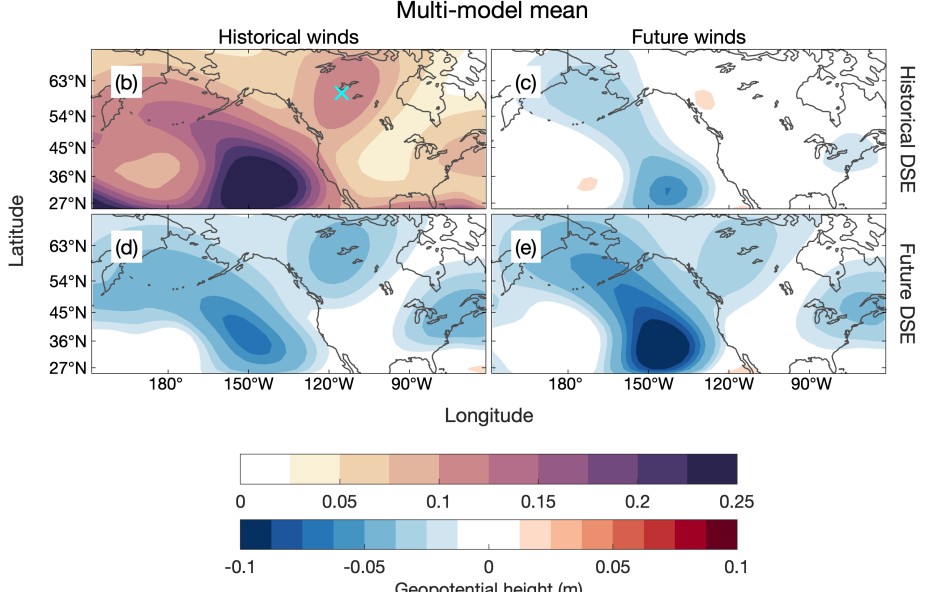

**Figure 2.** (a) The time series of geopotential height at 500 hPa at 60° N, 115° W for each January and their mean (thick black line) for the set of LBM simulations with the historical mean state winds and dry static energy (DSE) from ACCESS-ESM1-5. (b) The multi-model mean teleconnection amplitude (see text for definition) for the set of simulations with mean states given by historical winds and DSE. (c) The difference between the multi-model mean teleconnection amplitudes for the set of simulations with the full historical mean state and the set of simulations with future winds and historical dry static energy, (d) historical winds and future fry static energy, and (e) future winds and future dry static energy. DSE = dry static energy.

the change in the tropical vertical temperature gradient for individual CMIP6 models between the historical and future periods, confirms that models show a robust increase in the tropical dry static stability.

We now explore the inter-model spread of changes to the MJO teleconnection due to changes in the mean state. Figure 4 summarizes, for the separate mean states from each CMIP6 model, the change in the LBM-simulated teleconnection amplitude over the North Pacific (top panels; 150°-235° E, 25°-70° N) and North America (bottom panels; 25°-70° N and 235°-290° E, 25°-70° N), as a percent change per Kelvin multi-model mean warming. Here and throughout the remainder of this work,

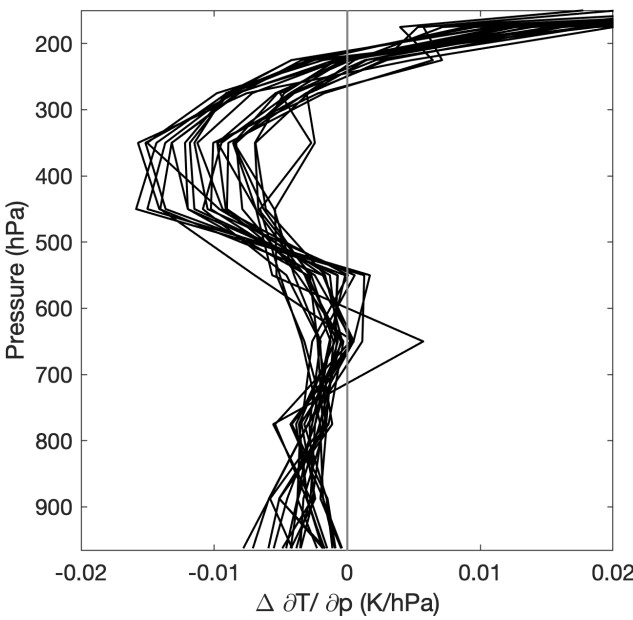

**Figure 3.** The change in the tropical mean (20°S-20°N) vertical temperature gradient ($\partial T/\partial p$) between the historical and future periods for each CMIP6 model used in this study.

we choose to separate the larger Pacific-North America region into two smaller regions because we do not want to assume that the amplitude changes in the two regions are the same. For each region we plot the domain-mean percent change in the teleconnection amplitude, as well as the fractional area that is strengthening for perturbations to *either* the mean state dry static energy or the mean state wind. By fractional area, we mean the fraction of the region that shows a teleconnection amplitude change greater than zero. We include the fractional area as a crude way to communicate the spatial distribution of teleconnection changes over the region, where a value of 0.5 indicates that 50% of the domain shows stronger teleconnections in the future period compared to the historical period. Likewise, a value of 1 indicates that the entire domain shows a larger teleconnection amplitude in the future period. In isolation, the mean teleconnection amplitude change over the region may be misleading if one small region experiences a very strong change in one direction (enough to make the domain mean change be of the same sign), despite the majority of the domain experiencing a change of the opposite direction. For example, we find that there are some cases in which the mean teleconnection amplitude change over the region is positive, while the majority *area* of the region shows a negative change.

The increase in the tropical dry static stability is a consequence of amplified upper-tropospheric warming that occurs as tropical temperatures adjust to warmer surface temperatures (e.g., Santer et al., 2005). Our first main finding is strong corroboration of the hypothesis that the increase of the tropical dry static stability is a robust thermodynamic response of the climate to warming that tends to reduce the MJO's teleconnections in almost all models (Fig. 4a,c). This is the first multi-model confirmation of this hypothesis, which first appears in the single-model study of Wolding et al. (2017). There is relatively good agreement in

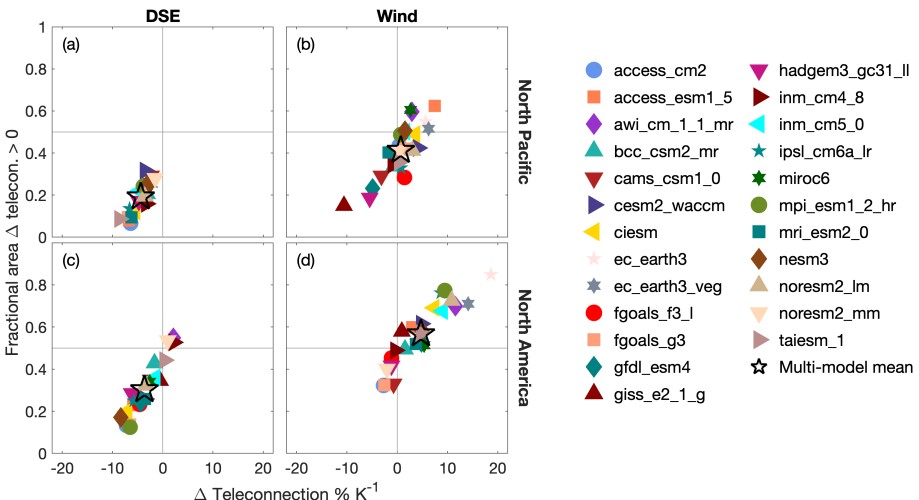

**Figure 4.** For individual CMIP6 model mean states and the multi-model mean, the change in the teleconnection amplitude (as a percent change per multi-model mean Kelvin warming) to the North Pacific (150°-235° E, 25°-70° N) and North America (235°-290° E, 25°-70° N) between LBM simulations with mean states given by future and historical quantities. The isolated change due to (a) stability, over the North Pacific; (b) wind, over the North Pacific; (c) stability, over North America; (d) wind, over North America. Horizontal axis is the domain mean change over the region, while the vertical axis shows the fractional area of teleconnection amplitude strengthening. DSE = dry static energy.

the teleconnection amplitude change due to the change in the atmosphere's thermal structure alone (i.e., neglecting mean state wind or MJO changes) between model mean states, particularly for teleconnections to the North Pacific. Comparison between panels a and c of Fig. 4 shows that there is greater spread downstream of the tropical heat source over North America, although reasons for this are unclear.

Despite agreeing on effects of stability, models disagree wildly on the effects of future wind changes. Figure 4b,d show the change in the teleconnection amplitude due to the change in the wind between the historical and future period. The multi-model mean response of the MJO teleconnection amplitude over the North Pacific is about zero, with a majority of the region experiencing a weakening. However, the inter-model spread is relatively large, with models showing changes between $-11$ to $7 \% \text{ K}^{-1}$ multi-model mean warming. Over North America, the multi-model mean change in the teleconnection due to the wind is a strengthening of $4.7 \% \text{ K}^{-1}$. But again, the inter-model spread is large ($-3$ to $19 \% \text{ K}^{-1}$).

In summary, three key results emerge concerning how mean state changes in a future warmer climate may contribute to changing MJO teleconnections to the North Pacific and North America. First, we find robust support for the expectation that increases in tropical dry static stability will contribute to weaker MJO teleconnections. Second, we find that changes to the mean state winds can contribute to very large changes in the MJO teleconnection; that is, large enough to as much as double or negate the weakening that is expected due to increasing tropical dry static stability. Third, there is a large amount of inter-model spread in LBM-simulated changes to the MJO teleconnection that result from changes to mean state winds, which suggests

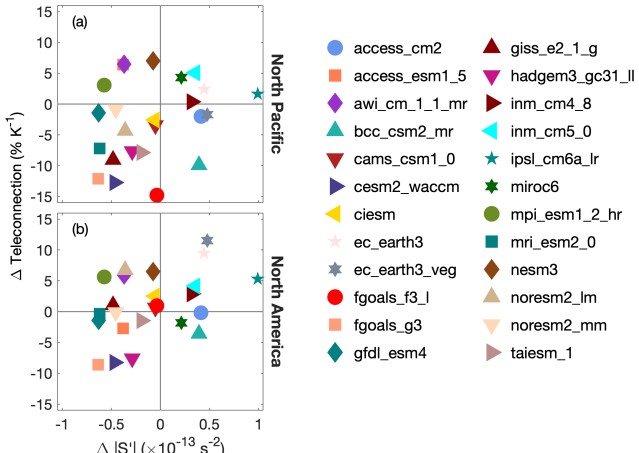

**Figure 5.** For individual CMIP6 model mean states, the difference in the Rossby wave source ($S'$) versus the difference in the teleconnection amplitude between simulations with historical and future mean state winds for (a) the North Pacific and (b) North America. The horizontal axis is the ensemble-mean, time-mean difference of the absolute value of $S'$ averaged over the subtropical jet (80°-190° E, 20°-40° N).

that uncertainty in the response of the mean state winds to warming is a large contributor to uncertainty in modeled changes to the MJO teleconnection.

It is thus important to better understand the cause of teleconnection changes associated with mean state wind changes in the
290 CMIP6 models. Mean state winds can increase or decrease the strength of MJO teleconnections via an intensification of Rossby wave excitation and/or through mean state changes that permit more Rossby wave energy propagation into these regions.

We begin by quantifying the strength of Rossby wave excitation using the linearized equation for the "Rossby wave source" (Sardeshmukh and Hoskins, 1988), $S'$, a term that represents the generation of large-scale vorticity,

$$S' = - \left[ \nabla \cdot (\overline{\mathbf{v}_\chi} \zeta') + \nabla \cdot (\mathbf{v}_\chi' \overline{\zeta}) \right] \tag{2}$$

where $\zeta$ is the absolute vorticity, $\mathbf{v}_\chi$ is the horizontal divergent wind, the overbar represents mean state quantities, and the prime represents anomalies. Here, anomalies are the LBM-simulated deviations from the user-input mean state. In the subsequent analysis, we use quantities at 200 hPa to calculate $S'$.

We do not find a systematic relationship between the difference in $S'$ and the difference in the mean teleconnection amplitude over the North Pacific or North America between simulations with historical and future winds. Figure 5 shows, for
simulations with historical and future winds, the relationship between the ensemble mean, time mean, difference in the absolute value of the Rossby wave source averaged over the subtropical jet (80°-190° E, 20°-40° N), and the difference in the mean teleconnection amplitude over the North Pacific and North America. Figure A3 in Appendix A verifies that this region indeed encloses the region of highest $S'$ variability in the LBM simulations. Models with larger increases in $|S'|$ over the jet do not systematically have stronger teleconnections. If changes to Rossby wave excitation were to play a key role in the difference in

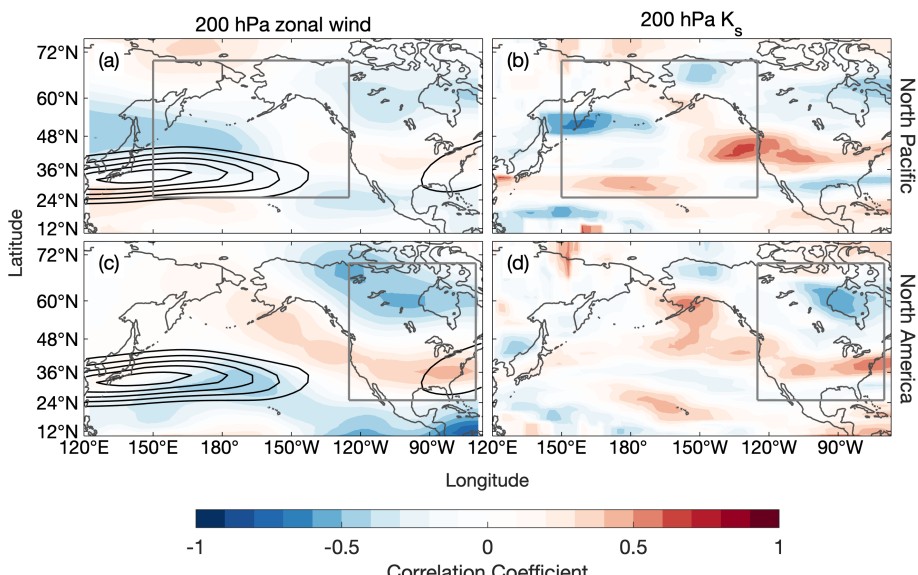

**Figure 6.** Correlation coefficient between the local change in the 200 hPa zonal wind and the regional mean change in the teleconnection amplitude over (a) the North Pacific and (c) North America. (b) Correlation coefficient between the local change in the stationary Rossby wave number, $K_s$, and the regional mean change in the teleconnection amplitude over the North Pacific and (d) North America. Grey boxes outline the North Pacific and North America regions.

the teleconnection amplitude between simulations with historical and future mean state winds, then it would be reasonable to expect either weaker or similar-sized links over North America and the North Pacific, since most Rossby waves excited by the MJO propagate first over the Pacific before reaching North America. Instead, the relationship between the difference in $|S'|$ and the teleconnection amplitude difference appears stronger over North America (r = 0.43) than over the North Pacific (r = 0.27). We therefore conclude that the change in $|S'|$ is not a primary mechanism that explains how mean state winds affect the

teleconnection amplitude difference in our LBM simulations. This is consistent with Wang et al. (2020) and Zheng and Chang (2020) who show that a stronger $S'$ does not necessarily lead to stronger MJO teleconnections.

   In addition to playing a key role in Rossby wave excitation, mean state winds also determine the path that Rossby waves take as they propagate (e.g., Karoly, 1983; Hoskins and Ambrizzi, 1993). Meridional shifts and variations in the zonal extent of the subtropical jet have been linked with variations in teleconnection amplitude across North America (Henderson et al.,

2017; Tseng et al., 2020b; Zheng and Chang, 2020). An eastward extension of the subtropical jet was shown to be important for increases in the MJO's impact over the North American west coast in climate models (Zhou et al., 2020). Similarly, Samarasinghe et al. (2021) propose that the changes to the North Pacific jet may explain modeled increases in simulated MJO teleconnections in CESM2-WACCM. Here, we search for a systematic (common across CMIP6 model mean states) and localized (specific to one region) change in the upper-tropospheric extratropical mean flow that explains variations in

the teleconnection amplitude (inter-model spread in Figure 4b,d). We begin by looking for a link between the change in the

extratropical zonal wind at 200 hPa and the teleconnection amplitude response to changes to the winds. Figure 6a,c shows the correlation coefficient between the local change in the 200 hPa zonal wind (the difference in the January mean zonal wind between the historical and future periods) and the regional mean (i.e., across the entire North Pacific or North America region) difference in the MJO teleconnection amplitude between simulations with historical and future mean winds (individual modeled changes in 200 hPa zonal wind are shown in Figure A4 in Appendix A). Correlations are generally weak everywhere (for reference, r = 0.5 corresponds to a p-value of roughly = 0.30 using a Fisher Z-transformation), which suggests that there may not be a change to the winds over a specific region that explains a majority of the spread in Figure 4b,d. However, given the importance of horizontal gradients of mean state vorticity for Rossby wave propagation, such a link may require more work to unveil.

Analyses of the stationary Rossby wave number have been useful for assessing how the mean state winds affect Rossby wave propagation (e.g., Henderson et al., 2017; Karoly, 1983; Tseng et al., 2020b; Wang et al., 2020; Zheng and Chang, 2020). The stationary Rossby wave number on a Mercator projection (to account for spherical geometry) for a zonal flow is defined as

$$K_s = a \left( \frac{\beta_M}{u_M} \right)^{1/2}, \tag{3}$$

where $\beta_M$ is the meridional gradient of absolute vorticity on a Mercator projection,

$$\beta_M = \frac{2\Omega \cos^2 \theta}{a} - \frac{\partial}{\partial y} \left[ \frac{1}{\cos^2 \theta} \frac{\partial}{\partial y} (u_M \cos^2 \theta) \right], \tag{4}$$

and $u_M = u/\cos\theta$ is the temporal mean, full (including summetric and asymmetric components) zonal wind divided by the cosine of latitude ($\theta$). In Eqs. (3) and (4), $a$ is the earth's radius and $\Omega$ is the earth's rotation rate. Hoskins and Ambrizzi (1993) showed that $K_s$ can be used to understand Rossby wave propagation: waves turn towards regions of higher $K_s$, are generally reflected away from regions where the zonal Rossby wavelength is equal to $K_s$ or where $\beta_M \leq 0$, and are either dissipated beyond or reflected from regions where $u \leq 0$. Regions of $K_s$ maxima, which tend to occur in strong westerly jets, act as Rossby waveguides. Multiple previous studies have used the stationary Rossby wave number in this form (i.e., neglecting meridional wind) when exploring the impact of the mean state winds on MJO teleconnections (Henderson et al., 2017; Tseng et al., 2020b; Wang et al., 2020; Zheng and Chang, 2020).

We now inspect if localized changes to $K_s$ at 200 hPa in the CMIP6 model January basic states can systematically explain the spread in Figure 4b,d. An increase in the local $K_s$ indicates that Rossby waves with shorter wavelengths are able to propagate further before turning. Similarly, a decrease in the local $K_s$ means that propagating Rossby waves will turn sooner. A change to the meridional width of the waveguide, which presents as an increase or decrease in $K_s$ along the north and/or south flanks of the waveguide, affects the trajectory of Rossby waves traveling within the waveguide, with Rossby waves able to travel further meridionally and turn less often in a broader waveguide (Hoskins and Ambrizzi, 1993). Thus, the path that Rossby waves take as they exit the jet is sensitive to both the width of the waveguide and the value of $K_s$. Figure 6b,d show the correlation coefficient

between the changes in the local $K_s$ and the regional mean teleconnection amplitude change for the North Pacific and North America, respectively (for reference, we include the difference in $K_s$ between the historical and future period for each model in Appendix A). We include boxes outlining the North Pacific and North America regions for reference. In the calculation of the correlation, we use averages of the local $K_s$ change over $10° \times 10°$ boxes to focus on the larger features; although using boxes of different sizes (we tried boxes ranging in size from $3°$-$20°$) does not qualitatively change the results. Regions of stronger positive correlation would be suggestive of a systematic (common across models) positive relationship between the local change in $K_s$ and the regional (i.e., across North Pacific or North America) teleconnection amplitude response to the changes in the winds projected by CMIP6. Similarly, regions of strong negative correlation would suggest a systematic inverse relationship between the local change in $K_s$ and the regional teleconnection change. Figure 6b,d shows that mostly, correlations are weak (p-values > 0.2 everywhere). We thus do not identify a clear systematic and localized mechanism that may explain how mean state wind changes lead to changes to teleconnectivity over the North Pacific or North America across CMIP6 models. More work is needed to understand how changes to the mean flow will impact future MJO teleconnections, which our results indicate are likely a large source of inter-model spread in future projections of MJO teleconnections. For example, consideration of the meridional wind as in Li et al. (2015), which is omitted in the form of the stationary Rossby wave number that we use here, may reveal mechanisms due to changes to the meridional, rather than the zonal wind. We leave this for future work.

### 3.2 MJO intensity and propagation characteristics

We now investigate how changes to MJO intensity and propagation characteristics affect the MJO teleconnection by conducting simulations in which the mean state is held constant, and either the propagation speed, eastward propagation extent, zonal wavenumber, or intensity of the idealized thermal forcing are perturbed to upper and lower bounds of expected end-of-century changes to the MJO. Instead of using mean states from all CMIP6 models, we use a subset of mean states from 10 models to minimize utilization of computational resources. For these experiments, we chose CMIP6 models that produced a minimum number of unstable ensemble members in experiments used in Sect. 3.1 (see Fig. A2).

Figure 7 summarizes the results of these simulations. As in Figure 4, the horizontal axis shows the regional mean difference in the teleconnection amplitude while the vertical axis shows the fractional area of the region that has stronger teleconnections (perturbed MJO compared to control MJO). Panels a and e show the change to the teleconnection amplitude over the North Pacific and North America, respectively, that result from perturbations to the magnitude of the propagating thermal forcing. The linearity of the LBM ensures that, in the absence of instability, the magnitude of the simulated response varies linearly with the magnitude of the forcing. We verified this with simulations using a 20% increase in the magnitude of the forcing for two different model basic states. For the remainder of the 8 models used in these experiments, we thus did not run simulations (see Table 1), because the magnitude of the extratropical response to the perturbed forcing amplitude is independent of the mean state. If nonlinear interactions had been considered, variations across the different model climates could have also been important in the response of the teleconnection to changes in heating intensity.

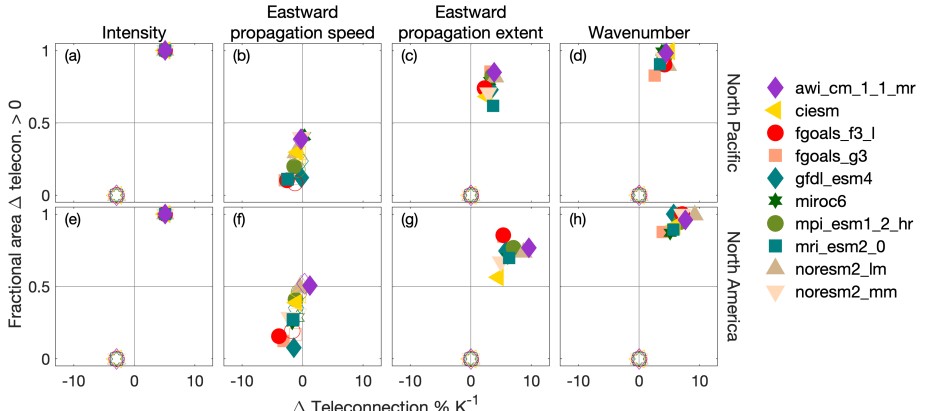

**Figure 7.** For individual CMIP6 model mean states, the change in the teleconnection amplitude (as a percent change per multi-model mean Kelvin warming) to the North Pacific (150°-235° E, 25°-70° N) and North America (235°-290° E, 25°-70° N) between LBM simulations with the control and perturbed thermal forcing. (a),(e) Response to perturbed heating intensity; (b),(f), eastward propagation speed; (c),(g), eastward propagation extent; and (d),(h) zonal wavenumber. Unfilled (filled) markers represent the lower (upper) bounds of perturbations to each MJO feature expected at the end of the century given a high emissions scenario (see Table 1).

Unlike perturbations to the heating intensity, the responses of the MJO teleconnection to perturbations to the eastward propagation speed (Fig. 7b,f), extent (Fig. 7c,g), and zonal wavenumber (Fig. 7d,h) are sensitive to the mean climate of each CMIP6 model. Consistent with previous work finding decreased MJO teleconnection strength with increased MJO propagation speed (Zheng and Chang, 2019; Wang et al., 2020; Bladé and Hartmann, 1995), we find that increases to the propagation speed of our idealized heating produce modest decreases in the teleconnection amplitude over the North Pacific and North America.

Increasing the eastward extent and decreasing the zonal wavenumber (i.e., increasing the zonal scale) of the propagating MJO thermal forcing in the LBM simulations increases the teleconnection amplitude for all model climates used and has the potential to produce the largest increases in the MJO teleconnection amplitude, more so over the North America region. For increases in the eastward propagation extent, this is likely related to an eastward broadening of the region over which the MJO excites Rossby waves. Adames and Wallace (2014) found that for the mean state winds of the period from 1979-2011, the MJO excites

the strongest extratropical response when heating maxima are located over the central Pacific. While we separately tested the sensitivity of the extratropical response to increases in the MJO's propagation speed and decreases to the zonal wavenumber, these two changes to the MJO are linked and will likely occur together (Rushley et al., 2019). These two changes to the MJO have opposite impacts on MJO teleconnection amplitude, with the net impact appearing to be a slight strengthening from the larger strengthening effect from the decreased zonal wavenumber.

For perturbations to the MJO's propagation characteristics, Fig. 7 highlights the sensitivity of the MJO teleconnection change to the mean climate of each CMIP6 model. That is, for a given perturbation to the MJO's propagation speed, zonal wavenumber, or eastward extent, inter-model differences in the mean climate lead to variations in the teleconnection change. Thus, not only are changes in the mean climate important for understanding how the extratropical MJO teleconnection will evolve for

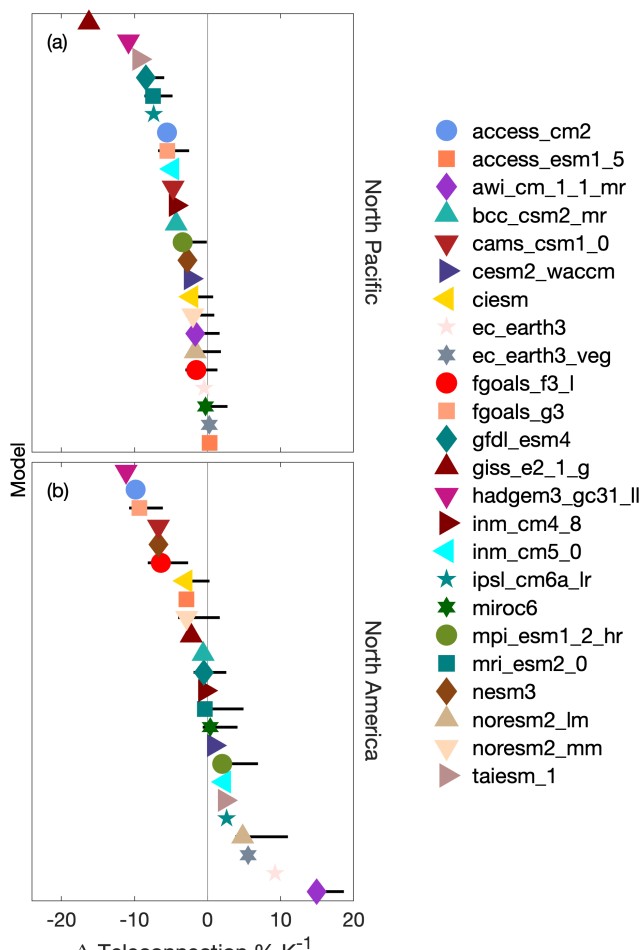

**Figure 8.** The total change to the MJO teleconnection over (a) the North Pacific and (b) North America due to the change in the full mean state (markers), and the potential range that results from considering changes to MJO propagation and intensity characteristics (horizontal lines extending from each marker) for the 10 models for which we ran perturbed MJO experiments.

an unchanging MJO, but also for understanding how MJO teleconnections will evolve in response to changes to the MJO. Additionally, for each individual MJO propagation or intensity characteristic, the spread in the MJO teleconnection change resulting from a perturbed MJO is on the order of the inter-model uncertainty that results from perturbations to the mean state alone (Fig. 4). These results challenge the notion that it may be possible to currently have confidence in future projections of MJO teleconnections.

### 3.3 Sum of mean state and MJO

Putting it together, we finish by quantifying the overall change in the MJO teleconnection that results when considering the linear sum of individual changes to the mean state and MJO features. For each CMIP6 model, Fig. 8 shows how the change

to the full mean state influences the change in the MJO teleconnection to the North Pacific and North America, as well as the the range of teleconnection changes that results from the sum of changes to the mean state and MJO propagation and intensity characteristics for the 10 CMIP6 mean states for which we also ran perturbed thermal forcing simulations. Over the North
Pacific, for all CMIP6 models, changes to the mean state alone lead to a weaker teleconnection for almost all models, with only two models (AWI-CM-1-1-MR, EC-EARTH3-VEG) showing a small strengthening. Factoring in changes to the MJO leads to a range of potential changes for many models that straddles zero. These results suggest that over the North Pacific, mean state changes may lead to weaker MJO teleconnections. However, a strengthening of the MJO teleconnection over the North Pacific, due to changes to the MJO (from potential increases in eastward propagation extent or heating intensity and decreases in zonal
wavenumber), can not be ruled out.

Over North America, there is no agreement on how changes in the mean state alone will impact MJO teleconnections: while the median change is near-zero, the range of MJO teleconnection changes due to mean state changes spans from -11 % $K^{-1}$ to 15 % $K^{-1}$. When changes to the teleconnection due to changes to the MJO are also considered, the final picture shows increased potential for strengthening, again due to the potential for an increase in the MJO's heating intensity and eastward
propagation extent, and decrease in its zonal wavenumber (Fig. 7).

## 4    Conclusions

The Madden-Julian Oscillation (MJO) is a key source of predictability of extratropical weather on subseasonal-to-seasonal timescales. While it is expected that the MJO teleconnection to the extratropics will change in a future warmer climate, models do not agree over much of North America (Zhou et al., 2020). Speculatively, this may be due in part to mean state biases or due
to inter-model spread or biases in the simulation of the MJO, both of which have been shown to lead to biases in the simulated MJO teleconnection of the current climate (Henderson et al., 2017; Wang et al., 2020).

The MJO teleconnection is sensitive to a number of factors, including the large-scale wind and thermal structure of the atmosphere and MJO propagation and intensity characteristics. While a number of climate models simulate the MJO and its teleconnections, each model represents a unique combination of multiple changes to the various features of the climate
that impact MJO teleconnections. This makes the determination of causal mechanisms important for the change to the MJO teleconnection a challenge. Additionally, many climate models do not accurately reproduce observed characteristics of the MJO or the mean state, and thus may not be reliable for studying internally simulated MJO teleconnections. Previous studies analyzing MJO teleconnections in climate models often analyze a subset of models that produce better MJOs (e.g., Henderson et al., 2017; Zhou et al., 2020).
Motivated by a desire to both (1) untangle the impacts of individual changes to features of the climate on changes to the MJO teleconnection with warming, and (2) maximize the resources offered by the most recent phase of the Coupled Model Intercomparison Project (CMIP6), we conduct a set of sensitivity experiments with a linear baroclinic model (LBM) of the atmosphere in concert with output from CMIP6. In each LBM experiment, a climate feature (i.e., the mean state or intensity or propagation characteristics of a propagating thermal forcing) is perturbed by a climate change amount; that is, the change

that is simulated in climate models by the end of the 21st century in simulations with strong $CO_2$ forcing. Specifically, we test sensitivity of the boreal winter MJO teleconnection amplitude over the North Pacific and North America to the mean state static stability and winds, and MJO eastward propagation extent, propagation speed, zonal wavenumber, and heating intensity.

    As expected, given that the increase of the tropical dry static stability is a robust thermodynamic response to surface warming, we verify that the increase in the tropical dry static stability with warming produces robust decreases in the LBM-simulated

MJO teleconnection amplitude. On the other hand, we find relatively large spread in the LBM-simulated teleconnection response to changes in the mean state wind. Nonetheless, when the mean state changes are considered together (Fig. 8), they lead to robust decreases in the LBM-simulated MJO teleconnection to the North Pacific. Over North America, the sum of mean state changes leads to a weaker/stronger MJO teleconnection for about half of CMIP6 model mean states.

    Investigation into a potential link between changes to the wind and changes to MJO teleconnection revealed no systematic

connection related to changes in the Rossby wave source. Additionally, we find no systematic and localized connection related to Rossby wave propagation over the extratropics. Thus, while future changes to the mean state winds have the potential to lead to large changes in MJO teleconnection amplitude over the North Pacific and North America, there does not appear to be a specific change to the zonal wind over a particular region that explains inter-model variations.

    We find that for perturbations to MJO characteristics (i.e., its intensity, eastward extent, zonal wavenumber, or propagation

speed), the uncertainty in the LBM-simulated MJO teleconnection that results is on the order of the uncertainty from the mean state changes alone. In particular, increases in the MJO's heating intensity, eastward extent, and zonal wavenumber have the potential to lead to large increases in the MJO teleconnection, with the sensitivity to propagation extent and zonal wavenumber being especially strong for the teleconnection for some model climates.

    Overall, over the North Pacific, despite a relatively large range in total teleconnection amplitude change of $-16$ to $0.3$ %

$K^{-1}$, which depends on the CMIP6 model used in the LBM simulations, in general LBM simulations suggest that the MJO teleconnection to the North Pacific may weaken with warming. Over North America, the picture is more blurry, with a large range of potential changes (between $-11$ to $15$ % $K^{-1}$) and many models also hovering near, or just above zero.

    In a multi-model study assessing internally-simulated MJO teleconnections in both CMIP5 and CMIP6, Zhou et al. (2020) find no relationship between changes to the MJO's circulation strength and teleconnection amplitude change over North Amer-

ica. The MJO circulation strength is primarily a function of the MJO's heating intensity and the tropical dry static stability (Wolding et al., 2016). Thus, the lack of a relationship between modeled MJO circulation change and teleconnection amplitude change may be due to overwhelming and independent model spread in other features of the climate system, namely the mean state wind, or propagation features of the MJO, such as its eastward extent, propagation speed, or zonal wavenumber. Our results support the hypothesis suggested in Samarasinghe et al. (2021), that the lack of an apparent relationship between

changes to MJO circulation strength and MJO teleconnection amplitude found in previous work appears to be due, in part, to large inter-model spread in future projections of the mean state winds.

    Over the North Pacific in CESM2-WACCM, Samarasinghe et al. (2021) find large increases in the internally generated MJO teleconnection strength between the pre-industrial period and the end of the century. They hypothesize that this strengthening, which appears despite the weakening expected from increases in tropical dry static stability, may be due to changes in the mean

winds that enhance wave propagation over the region. Here, we find that mean state changes alone in CESM2-WACCM lead to a modest *weakening* over the region (Fig. 8a). Thus, it is possible that the strengthening MJO teleconnection over the North Pacific simulated directly in CESM2-WACCM is due to changes in the MJO. In addition, inter-model spread in features of the MJO in CMIP6 may also be contributing to the results found in Zhou et al. (2020), which we leave for future work.

We have thus identified sources of uncertainty leading to uncertainty in projections of MJO teleconnections in a future warmer climate. To first order, LBM simulations show that model spread in projections of future winds is the biggest source of uncertainty in projecting the response of MJO teleconnections to warming. Hence, increased confidence in changes to the extratropical mean flow may help reduce uncertainty in teleconnection changes over the North Pacific and North America. Additionally, uncertainty in the mean climate also contributes to spread in how MJO teleconnections respond to some changes to the MJO. Lastly, uncertainty in how the MJO will respond to climate change is nontrivial, which is complicated further by the fact that many models still struggle to produce realistic MJOs (e.g., Ahn et al., 2020).

Over North America, a possible future outcome is that of the median result found in Fig. 8 of a near-zero regional mean teleconnection change. In this case, it is possible that while the mean change over the region as a whole is zero, that the pattern of the teleconnection may change, leading to increases in the teleconnection amplitude over some parts of North America and decreases elsewhere. In this case, a reduction of the uncertainty in future projections of the mean state winds will also help reduce uncertainty in the MJO teleconnection change, due to the tight dependence of the stationary wave pattern on the mean winds.

A discussion of the caveats and limitations of our study is necessary. Firstly, we readily admit the limitations of our choice of a linear model to explore how the MJO teleconnection responds to changes in the mean state and to the MJO. While this allows us to easily diagnose causal relationships, span a previously sparsely sampled causative parameter space, and separately diagnose the effect of changes to mean state winds versus the dry static stability on MJO teleconnections using multiple model mean states; the linear framework is missing nonlinear relationships between the mean state, the MJO forcing, and MJO-excited Rossby waves. For example, the mean state exerts a strong control on MJO propagation characteristics (Jiang et al., 2020), which we have not included here. Rossby waves excited by the MJO are also able to extract energy from the mean flow (Adames and Wallace, 2014; Zheng and Chang, 2020), and nonlinear interactions can cause a spatial shift in extratropical maximum geopotential height anomalies associated with the MJO (Lin and Brunet, 2018). Both eddy-mean flow interactions, and any MJO dependence on the mean state are not simulated in the LBM. While previous work has shown that the MJO teleconnection is, to first order, linear (Mori and Watanabe, 2008; Lin and Brunet, 2018), future work involving a nonlinear framework may be needed to reduce uncertainty in future projections of the MJO teleconnection to the extratropics.

Secondly, our results are sensitive to the ideal forcing used to explore how a changing MJO may affect teleconnection changes. We have used a very idealized heating to represent the MJO which is constant across LBM simulations. Thus, despite using a range of perturbations of this heating to sample the range of possible changes to the MJO and subsequent impacts on teleconnections, we are lacking a comprehensive sampling of the inter-model variations in future MJO characteristics. Perturbations rely on projected changes from CMIP5, rather than CMIP6 (the latter of which has an improved MJO (Ahn et al., 2020)) due to the availability of detailed published analyses of modeled changes to the MJO. We also examine only a subset

of the possible changes to the MJO that may occur; namely, its heating intensity, zonal wavenumber, propagation speed, and eastward propagation extent. For example, we have not investigated how changes to the MJO's frequency, meridional extent, or other changes to its spatial structure may affect future MJO teleconnections. However, given that many models still struggle to reproduce a reliable MJO in the current climate, our current understanding of how the MJO may change with warming is incomplete. Thus there is much more work that needs to be done to understand how MJO teleconnections may change in the

future.

Lastly, we have considered only January mean states for this work. It is possible that results could be different for other months or seasons. For example, changes in the mean wind may be less important during the shoulder months when the East Asian subtropical jet is weaker.

For now, we suggest this work as a compelling first step towards understanding the intriguing model spread in future MJO

teleconnection strength in the extratropics, and another reason to elevate the ongoing community quest to reduce uncertainty of the mean midlatitude circulation's response to climate change. Work comparing these results with changes to the MJO teleconnection directly simulated by CMIP6 models, while outside the scope of the present work, will additionally help deepen understanding of how MJO teleconnections may change in the future and are promising topics for future work.

*Code and data availability.* Data to reproduce figures is published online at https://zenodo.org/ with doi:10.5281/zenodo.4737438. The lin-

ear baroclinic model is available through request from M. Hayashi (michiyah@hawaii.edu). CMIP6 data is publicly available through the Earth System Grid Federation at https://esgf-node.llnl.gov/search/cmip6/.

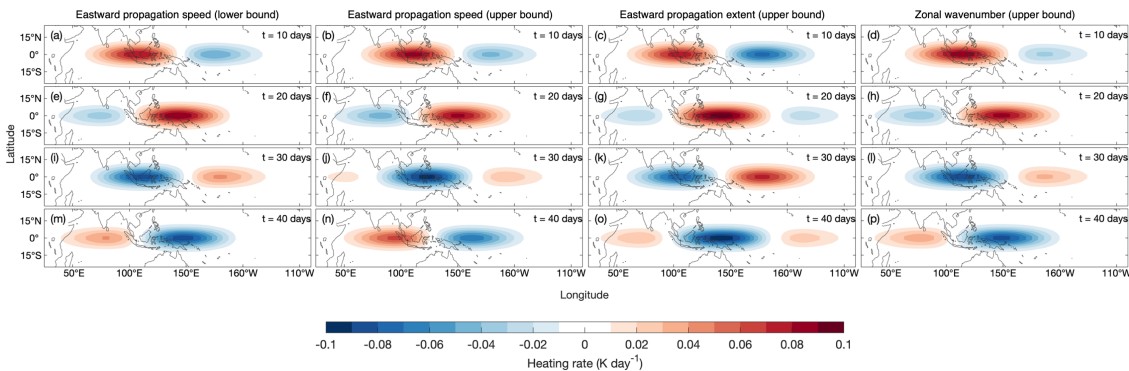

**Figure A1.** As in Figure 1, but for the perturbed MJO cases for (a,e,i,m) the lower bound of the propagation speed experiment; (b,f,j,n) the upper bound of the propagation speed experiment; (c,g,k,o) the upper bound of the increased eastward propagation extent experiment; and (d,h,l,p) the upper bound of the perturbed wavenumber experiment.

**Appendix A**

Figure A1 shows the idealized forcing at days 10, 20, 30, and 40 for the cases where the forcing is perturbed.

Figure A2 shows the number of simulations that became unstable (and hence omitted from our analysis; see Sect. 2), and indicates the CMIP6 models for which all simulations were excluded from our analysis for having 10 or more unstable ensemble members for a particular basic state combination.

Figure A3 shows the temporal variance of the ensemble mean Rossby wave source ($S'$) in the LBM for each CMIP6 model historical basic state used in the analysis. The box around the region (80°-190° E, 20°-40° N) indicates the region used for Fig. 5 of the main text.

Figure A4 shows the difference in the January mean 200 hPa zonal wind between the historical and future periods for individual CMIP6 models, with contours outlining the subtropical jet for reference.

Figure A5 shows the difference in the January mean $K_s$ at 200 hPa between the historical and future periods for the models included in the analysis. Contours outlining the subtropical jet are included for reference.

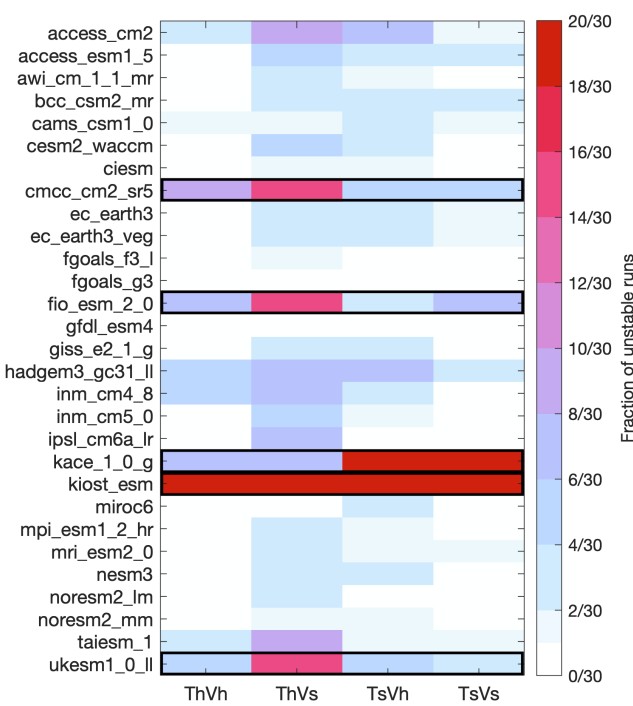

**Figure A2.** For each model basic state combination, the fraction of the 30 ensemble members that met the instability criterion described in Sect. 2. Models omitted from our analysis (those for which 10 or more of their ensemble members for any of the basic state combinations became unstable) are indicated by the black outline. Th (Ts) = basic state dry static stability from the historical (future) period; Vh (Vs) = basic state winds from the historical (future) period.

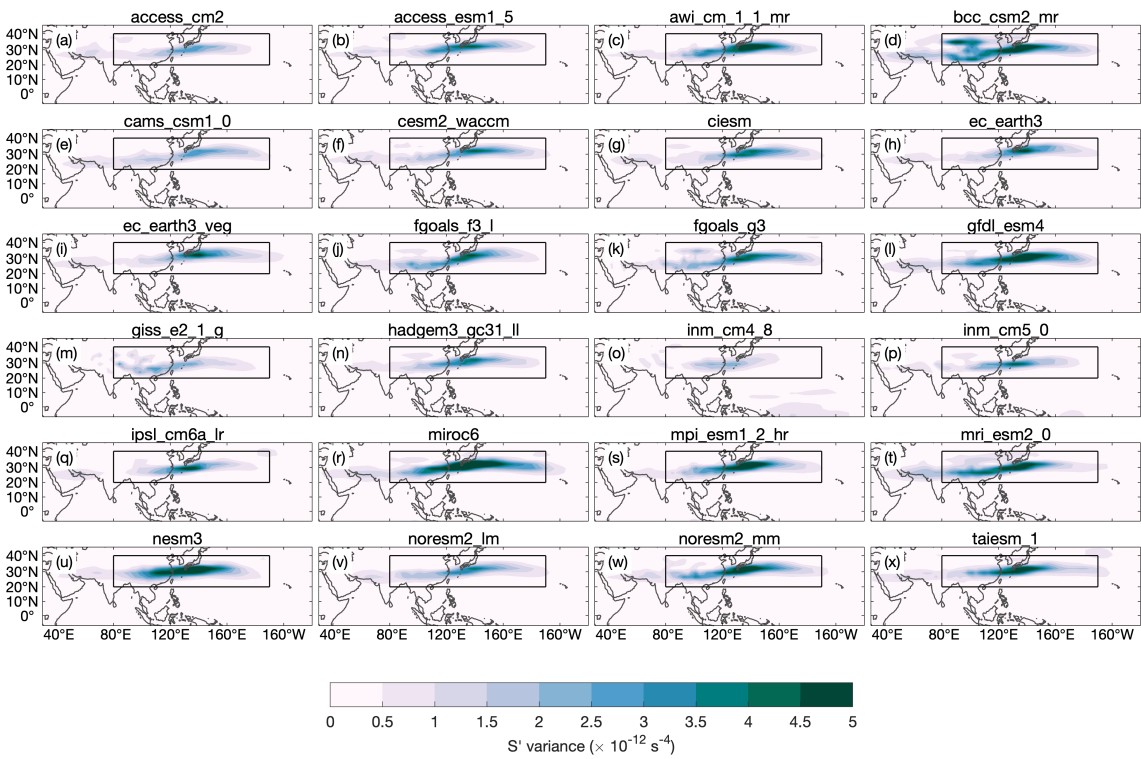

**Figure A3.** The temporal variance of the ensemble mean Rossby wave source ($S'$) with the full mean states given by historical quantities. The box around the region ($80°$-$190°$ E, $20°$-$40°$ N) indicates the region used for Figure 5 of the main text.

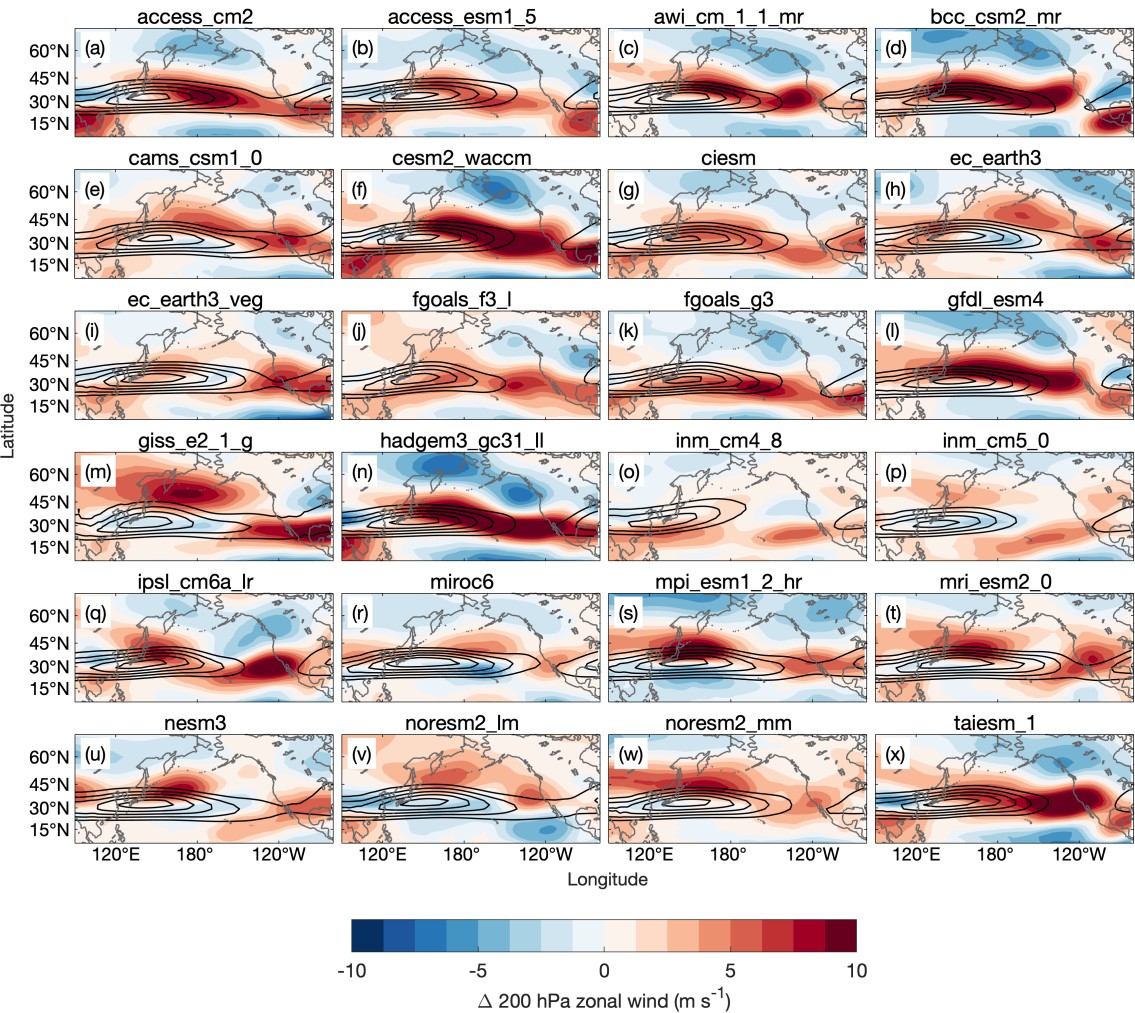

**Figure A4.** Shading is the difference in the January mean zonal wind at 200 hPa between the historical (1985-2014) and future periods (2071-2100) for models included in the analysis. Solid black contours show the January mean zonal wind at 200 hPa, starting at 35 m s$^{-1}$ and increasing by 10 m s$^{-1}$.

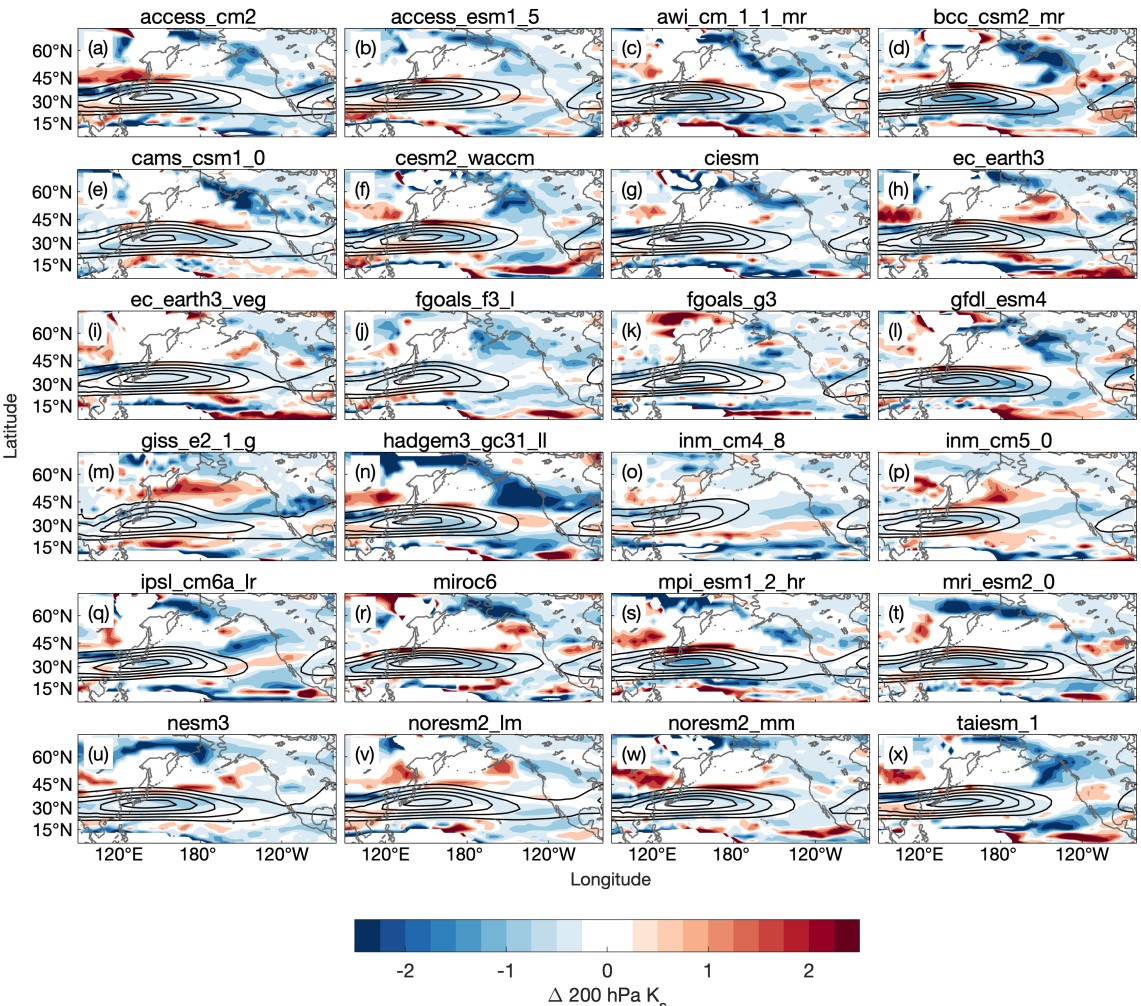

**Figure A5.** Shading is the difference in the January mean $K_s$ at 200 hPa between the historical (1985-2014) and future periods (2071-2100) for models included in the analysis. Solid black contours show the January mean zonal wind at 200 hPa, starting at 35 m s$^{-1}$ and increasing by 10 m s$^{-1}$.

*Author contributions.* AMJ led the design of the experiments, performed the simulations, and prepared the manuscript. DAR and EAB aided with the experimental design, interpretation of the results, and writing of the manuscript.

*Competing interests.* The authors declare no competing interests

*Acknowledgements.* We thank David Straus and two anonymous reviewers for helpful comments. We thank Michiya Hayashi for sending the LBM code, and acknowledge Donald Dazlich, Stephanie Henderson, and Kai-Chih Tseng for their help in getting the model set up and running. We also thank Michael Pritchard for help with manuscript editing and Eric Maloney and Tom Beucler for helpful suggestions. This work was supported by the National Science Foundation under award AGS-1826643, by the National Oceanic and Atmospheric Administration (NOAA) under award NA19OAR4590155, and by the NOAA Climate and Global Change Postdoctoral Fellowship Program, administered by UCAR's Cooperative Programs for the Advancement of Earth System Science (CPAESS) under award NA18NWS4620043B. Support for EAB was provided, in part, by NSF grant AGS-1749261.

This research also used computing resources of the Extreme Science and Engineering Discovery Environment (XSEDE), which is supported by National Science Foundation grand number ACI-1548562 (Towns et al., 2014) and allocation number TG-ATM190002.

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
