# Peer review of "Drivers of uncertainty in future projections of MJO teleconnections"

_Weather and Climate Dynamics, 2021_

## Author Response (AR1)

**Note**
Referee comments are in black. Author responses are below each comment in red.
* * *
**Referee #1: David Straus**

The paper presents a well thought-out and intelligent approach to determining the mechanisms behind changes in the strength of the Madden-Julian Oscillation (MJO) teleconnections in future climate simulations. In particular the paper explores the roles of the changes in basic state (including static stability and the basic state winds) as well as future changes in the characteristics of the MJO itself. The results also indicate the degree to which these mechanisms differ in the various CMIP 6 models. The results are fairly convincing. Subject to some revisions, the paper should be published.

There is one relatively major point I would like to bring up, namely the choice of the teleconnection metric. I should first say that I applaud the use of the temporal variance, since in contrast to many other MJO studies, this choice implies the recognition that the MJO forcing (and hence the wave-train response) propagates in a continuous fashion, and does not make the stationary-wave response hypothesis so often invoked. However, I am puzzled by the choice of the low-level v-wind field. The authors stater that: "We use low-level meridional wind in this calculation (instead of other dynamically relevant variables like mid-tropospheric geopotential height, for example), because the MJO primarily drives surface temperature variability across North America via low-level temperature advection …" This is taking a rather limiting point of view, appropriate if the North America Tsfc teleconnection were the main teleconnection of interest. But I think that the authors should (and do) take a wider point of view, for which a teleconnection metric involving perhaps the upper-level (300 hPa) temporal variance of height or v-wind would be more relevant, and would enable the authors to compare their results with the wealth of literature using upper-level fields to gauge the MJO teleconnection. Applying this metric would mean simply re-running the various codes the authors have on a different field, and the comparison between the results for the upper-level and low-level metrics would be quite illuminating.

We have repeated our analysis using geopotential height at 500 hPa (z500), with some key changed results (listed below). We also repeated the analysis using meridional wind at 300 hPa, as suggested, and the results were about the same as those we obtained with the commonly-used z500.
- Correlations between the change in the stationary Rossby wave number and the change in the teleconnection amplitude over North America interestingly disappeared.
- More models showed increases in the teleconnection strength over North America. The final result is about the same, however (near-zero median change).

There are a few conceptual points on which the authors need to expand further (if briefly). As described, the linear baroclinic model is damped enough to prevent the growth of baroclinic waves, which means that the role of the storm-track transient eddy convergences of heat and momentum are not taken into account explicitly. However, since the mean state u-wind configuration is taken from the CMIP6 simulations, the eddy-mean flow feedback is taken into account implicitly. This should mentioned in the discussion of the LBM.

We have included the following sentences directly after our discussion of the damping used in the LBM:

"As a consequence of the strong damping and linear framework, eddy convergences of heat and momentum in the extratropical storm tracks, including those associated with MJO variability \citep[e.g.,][]{Deng2011,Guo2017,Takahashi2014}, are not explicitly accounted for. However, because the mean state wind used in the LBM simulations is taken from CMIP6, eddy-mean flow interactions are, in a way, included implicitly. "

Secondly, Hoskins and Ambrizzi (1993) show that the waveguide is formed in areas near the maximum in Ks as a function of latitude, and has a width that extends between the points of inflection on either side of the maximum (see their Figure 2E). The width of the waveguide may play as large a role as the maximum value of Ks. The discussion of Figure 2 should be modified in this light.

We are assuming that the reviewer meant Figure 5 (which is about Ks) instead of Figure 2 (does not include any information about Ks).

In response to this comment, we have added the following discussion to the front of the paragraph describing Figure 5:

"We now inspect if localized changes to $K_s$ at 200 hPa in the CMIP6 model January basic states can systematically explain the spread in Figure \ref{fig:bs_scatter}b,d. An increase in the local $K_s$ indicates that Rossby waves with shorter wavelengths are able to propagate further before turning. Similarly, a decrease in the local $K_s$ means that propagating Rossby waves will turn sooner. A change to the meridional width of the waveguide, which presents as an increase or decrease in $K_s$ along the north and/or south flanks of the waveguide, affects the trajectory of Rossby waves traveling within the waveguide, with Rossby waves able to travel further meridionally and turn less often in a broader waveguide \citep{Hoskins1993}. Thus, the path that Rossby waves take as they exit the jet is sensitive to both the width of the waveguide and the value of $K_s$."

Minor Points:

(1) The paragraph regarding effect of propagation speed (lines 59-64) should refer to Yadav and Straus (2017) who discuss the observed teleconnections from observed fast and slow MJO episodes.

We have included the following sentence to the paragraph,

"Comparisons between composites of observed geopotential height anomalies during fast and slow MJO events show differences in both the teleconnection pattern and magnitude of the anomalies \citep{Yadav2017}."

(2) Figure 1 – is the "Heating rate" (y-axis) the heating amplitude A in equation 1?\

The contours show the perturbation heating the LBM is forced with (T' in equation 1). We have updated the text as follows (bold=new):

"Figure 1 shows the perturbation heating **(T')** in the mid-troposphere where it is a maximum for t= 10 days, 20 days, 30 days, and 40 days."

In responding to this comment, we found a small error in how equation 1 was written (dT'/dt was written instead of T'), which we have fixed.

(3) The definition of the teleconnection strength is confusing. In lines 183-184: "…calculate the teleconnection strength at each point as the square root of twice the variance of the ensemble mean meridional wind at 850 hPa" Presumably you mean twice the temporal variance (and not the ensemble spread). Also, do you take the ensemble mean first and then take the variance (as indicated in the text above), or take the variance of each run and then take the model mean (as indicated in the caption to Figure 2)?

The teleconnection strength is the square root of twice the variance of the ensemble mean: we take the ensemble mean first and then take the variance.

We have added the word "temporal" before the word "variance" as you have suggested in the text, and have added the following sentence directly after: "That is, we take the ensemble mean first and then calculate the variance." We have updated the figure caption so that it refers to the text (this particular sentence was confusing as written because in plotting Figure 2, we are taking the multi-model (CMIP6) mean of the "teleconnection amplitude" for each model.

(4) lines 209 – 213: the eastward extension of the teleconnections near California is not convincing from Figures 2c-2e.

We have removed this sentence.

(5) lines 239-241: is the multi-model mean teleconnection strength different from 0 in any meaningful way?

We now find a multi-model strengthening of the teleconnection over North America due to the wind.

(6) Rossby wave source: equation (2) – what is the definition of the "anomaly" for each model. Is this the difference between future and historical runs?

The LBM directly simulates and outputs "anomalies": i.e., any deviations from the user-input basic state. We have added the following sentence:

"Here, anomalies are the LBM-simulated deviations from the user-input mean state."

(7) line 303-305: Figure 5. This figure is not explained well. I am still not sure which is the index in this figure and which is the geographically varying field. I think the index is the area-averaged teleconnection amplitude and the varying field is Ks (albeit smoothed by a running 15 x 15 degree average). This needs to be clarified.

Since the change in Ks between historical and future runs is not shown, how do we interpret the results of Figure 5 in the context of the general statement made in lines 352-353: "Over North America, the change in the mean state alone also leads to decreases in the teleconnection amplitude for most CMIP6 models"?

How does Figure 5 explain (as stated in the conclusions): "For teleconnections to North America in particular, we have identified a region over the eastern Pacific where changes to the winds appear to explain much of the inter-model spread"?

Switching from using meridional wind at 850 hPa to geopotential height at 500 hPa erased the link we had previously found between changes to the stationary Rossby wave number over the Gulf of Alaska and the LBM-simulated response to future projections of the wind. Our results using meridional wind at 300 hPa are about the same as those we find using z500, which lends confidence to our new conclusions (that of not finding a systematic and localized link between changes to the winds and MJO teleconnection changes).

(8) Figure 6a,d : Does a change in fractional area of 1 mean a doubling in size of areas where teleconnections > 0 ?

The teleconnection amplitude metric is a positive definite quantity (teleconnection amplitude is positive everywhere). These figures are showing the fractional area, for each region, where the change in the teleconnection amplitude is greater than 0. This means that a value of 1 in Figure 6a,d means that teleconnections strengthen everywhere.

Where this type of figure is first introduced (where Figure 3 is first mentioned in the text), we include the following (bold=new),

"For each region we plot the domain-mean percent change in the teleconnection amplitude, as well as the fractional area that is strengthening. We include the fractional area as a crude way to communicate the spatial distribution of teleconnection changes over the region, **where a value of 0.5 indicates that 50\% of the domain shows stronger teleconnections in the future period compared to the historical period. Likewise, a value of 1 indicates that the entire domain shows a larger teleconnection amplitude in the future period.**"

Additionally, we include these sentences when Figure 6 is introduced in the text,

"As in Figure 3, the horizontal axis shows the regional mean difference in the teleconnection amplitude while the vertical axis shows the fractional area of the region that has stronger teleconnections (perturbed MJO compared to control MJO)."

(9) line 396: "Overall, over the North Pacific, despite a relatively large range of −21 to 0.6 % K−1…" The rest of the sentence never states what quantity is being referred to.

We update the sentence.

References:

Yadav, P. and D. M. Straus, 2017: Circulation Response to Fast and Slow MJO Episodes. Mon. Wea. Rev., 145, 1577-1596.

**Referee #2**

This study examines the uncertainty in MJO teleconnection amplitude changes in North Pacific and North America in January based on results from the linear baroclinic model (LBM). The model is forced with the CMIP6 mean state and an idealized MJO heating with multiple sensitivity runs to test the relative contributions of different MJO and mean state characteristics to the uncertainty in MJO teleconnection amplitude changes. The results indicate that future changes in the mean state wind largely lead to uncertainty in the teleconnection amplitude over both the North Pacific and North America regions. Changes in the MJO propagation speed also contributes to the teleconnection uncertainty. Although some comprehensive simulations are run, the analysis approach in this study may need justification/improvement. Some results are not convincing as they are based only on the LBM and lack other scientific support. I, therefore, recommend this manuscript go through a major revision based on comments below.

Major comments:

Some analysis method needs justification/modification. For example:

1.  L91-93. In terms of the mechanisms related to teleconnection amplitude, the LBM could be TOO simple. Teleconnection amplitude is largely influenced by nonlinearity (Lin and Brunet 2018). Some relationships that may not found in full GCMs such as weaker MJO circulation leads to weaker teleconnection amplitude (Zhou et al. 2020) can be found in the LBM because the LBM is linear. This will make some results unrealistic/less convincing. Therefore, the authors may need to compare their results from the LBM with CMIP6 to discuss which relationships may be due to the linear setting, and which are not. Another way to address this issue is that the LBM package also has a nonlinear version (used in Henderson et al. 2018) which may be better for the investigation of the teleconnection amplitude changes.

Zhou et al. (2020) find no relationship between changes in MJO circulation strength and MJO teleconnection amplitude. Changes to the MJO circulation strength are directly impacted by changes in both the precipitation/heating intensity and dry static stability (Wolding et al., 2016; Wolding et al., 2017). As you have alluded to, this is indeed puzzling and was a key motivation for this work. Our leading hypothesis for why this relationship did not appear in Zhou et al. (2020) is that the model spread in other factors (e.g.,, the mean state wind) overwhelms any relationship between the MJO circulation strength and the teleconnection amplitude. That is, our hypothesis going into this work was that the leading driver of changes to MJO teleconnections is not MJO circulation change but rather mean state *wind* changes. We believe that if there were a way for Zhou et al. (2020) to control for changes to the mean state wind (among other shifts to the MJO), then they would have been able to show a positive relationship between modeled change in MJO circulation and MJO teleconnection amplitude. However, there is not a straightforward way to do this using solely CMIP6 output, hence our study using a linear model.

In setting up this experiment, we desired a framework in which to separately diagnose the impact that changes to the mean state winds and changes to the atmospheric thermal structure (i.e., its stability) have on changes to the teleconnection. Doing so is impossible in a nonlinear framework because the winds *must* be in balance with the temperature at equilibrium. This is why we chose the LBM for our study.

Our results corroborate our hypothesis that the very large spread in future projections of extratropical winds are the leading driver of uncertainty in projections of future MJO teleconnections. We continue to hypothesize that Zhou et al. (2020) could not find the expected connection between MJO circulation change and MJO teleconnection amplitude change because the inter-model spread in extratropical winds (and the resulting impact on the MJO teleconnection) is much larger than the inter-model spread in MJO circulation change. We do not believe that the lack of an apparent relationship between MJO circulation changes and MJO amplitude changes found in Zhou et al., (2020) primarily results from nonlinearity.

We have added the following paragraphs to the conclusions,

"In a multi-model study assessing internally-simulated MJO teleconnections in both CMIP5 and CMIP6, \citet{Zhou2020} find no relationship between changes to the MJO's circulation strength and teleconnection amplitude change over North America. The MJO circulation strength is primarily a function of the MJO's heating intensity and the tropical dry static stability \citep{Wolding2016}. Thus, the lack of a relationship between modeled MJO circulation change and teleconnection amplitude change may be due to overwhelming and independent model spread in other features of the climate system, namely the mean state wind, or propagation features of the MJO, such as its eastward extent, propagation speed, or zonal wavenumber. Our results support the hypothesis suggested in \citet{Samarasinghe2021}, that the lack of an apparent relationship between changes to MJO circulation strength and MJO teleconnection amplitude found in previous work appears to be due, in part, to large inter-model spread in future projections of the mean state winds.

Over the North Pacific in CESM2-WACCM, \citet{Samarasinghe2021} find large increases in the internally generated MJO teleconnection strength between the pre-industrial period and the end of the century. They hypothesize that this strengthening, which appears despite the weakening expected from increases in tropical dry static stability, may be due to changes in the mean winds that enhance wave propagation over the region. Here, we find that mean state changes alone in CESM2-WACCM lead to a modest \textit{weakening} over the region (Fig. \ref{fig:sum}a). Thus, it is possible that the strengthening MJO teleconnection over the North Pacific simulated directly in CESM2-WACCM is due to changes in the MJO. In addition, inter-model spread in features of the MJO in CMIP6 may also be contributing to the results found in \citet{Zhou2020}, which we leave for future work. "

Nonetheless, as you have mentioned, some previous work suggests that nonlinearity can be important for MJO teleconnections (Lin and Brunet, 2018; Zheng and Chang, 2020). However, there is also a strong body of work providing support for the use of a linear framework to study MJO teleconnections and the LBM has been shown in many studies to reproduce the salient features of the MJO teleconnection (e.g., Henderson et al., 2017; Tseng et al., 2019). In fact, Lin and Brunet (2018) show that the MJO teleconnection is largely linear in the first week from the tropical heating, and that beyond 10 days, the linear response is both stronger and has stronger statistical significance than the nonlinear component of the teleconnection. We have added the following sentences to the conclusions,

"A discussion of the caveats and limitations of our study is necessary. Firstly, we readily admit the limitations of our choice of a linear model to explore how the MJO teleconnection responds to changes in the mean state and to the MJO. While this allows us to easily diagnose causal relationships, span a previously sparsely sampled causative parameter space, and separately diagnose the effect of changes to mean state winds versus the dry static stability on MJO teleconnections using multiple model mean states; the linear framework is missing nonlinear relationships between the mean state, the MJO forcing,

and MJO-excited Rossby waves. For example, the mean state exerts a strong control on MJO propagation characteristics \citep{Jiang2020}, which we have not included here. Rossby waves excited by the MJO are also able to extract energy from the mean flow \citep{Adames2014,Zheng2020}, and nonlinear interactions can cause a spatial shift in extratropical maximum geopotential height anomalies associated with the MJO \citep{Lin2018}. Both eddy-mean flow interactions, and any MJO dependence on the mean state are not simulated in the LBM. While previous work has shown that the MJO teleconnection is, to first order, linear \citep{Mori2008, Lin2018}, future work involving a nonlinear framework may be needed to reduce uncertainty in future projections of the MJO teleconnection to the extratropics."

Finally, while we agree that there is much more work to be done in regards to how the MJO teleconnection may change in a future warmer climate, including the reviewer's excellent suggestions to compare these results with those directly simulated within CMIP6 or to repeat this study using a nonlinear model, this work is unfortunately outside the scope of the present paper. Not to mention, there is already ongoing work looking at changes to MJO teleconnections simulated directly within CMIP6 models (see abstract A167-08 from AGU Fall Meeting 2020 by Jiabao Wang, "Future Changes in MJO Teleconnections and its Impacts on Precipitation Extremes").

We have added the following sentence to the conclusions,

"Work comparing these results with changes to the MJO teleconnection directly simulated by CMIP6 models, while outside the scope of the present work, will additionally help deepen understanding of how MJO teleconnections may change in the future and are promising topics for future work. "

Citations

Henderson, S. A., Maloney, E. D., & Son, S. W. (2017). Madden-Julian oscillation Pacific teleconnections: The impact of the basic state and MJO representation in general circulation models. Journal of Climate, 30(12), 4567–4587.

Lin, H., & Brunet, G. (2018). Extratropical Response to the MJO: Nonlinearity and Sensitivity to the Initial State. *Journal of the Atmospheric Sciences*, *75*(1), 219–234.

Tseng, K.-C., Maloney, E., & Barnes, E. (2019). The Consistency of MJO Teleconnection Patterns: An Explanation Using Linear Rossby Wave Theory. *Journal of Climate*, *32*(2), 531–548.

Wolding, B. O., Maloney, E. D., & Branson, M. (2016). Vertically resolved weak temperature gradient analysis of the Madden-Julian Oscillation in SP-CESM. *Journal of Advances in Modeling Earth Systems*, 8(4), 1586–1619.

Wolding, B. O., Maloney, E. D., Henderson, S., & Branson, M. (2017). Climate change and the Madden-Julian Oscillation: A vertically resolved weak temperature gradient analysis. *Journal of Advances in Modeling Earth Systems*, 9(1), 307–331.

Zheng, C., & Chang, E. K.-M. (2020). The Role of Extratropical Background Flow in Modulating the MJO Extratropical Response. *Journal of Climate*, *33*(11), 4513–4536.

Zhou, W., Yang, D., Xie, S.-P., & Ma, J. (2020). Amplified Madden–Julian oscillation impacts in the Pacific–North America region. *Nature Climate Change*. https://doi.org/10.1038/s41558-020-0814-0

2. The authors investigated changes in the MJO propagation speed, zonal extent, intensity, and mean state dry static stability and winds. It is not clear why only these changes are examined. Are the authors choosing them randomly or based on some studies that these are the most robust and significant changes in the MJO and basic state?

We chose to separate the total mean state into mean state winds (u,v) and mean state temperature (which we refer to as the mean state dry static stability, as the vertical gradient of temperature is the static stability), in order to separately diagnose the impact that changes in each have on changes to the teleconnection. We are now more clear in the methods about which variables are kept constant when testing sensitivity to the mean state.

Regarding specific changes to the MJO, we tried to be comprehensive: we chose precipitation intensity because it directly relates to the strength of the circulation (and hence the magnitude of Rossby wave excitation); propagation speed because previous studies have shown it to be important for teleconnections and because of previous work suggesting that the MJO will become faster with surface warming; and eastward extent because of previous work suggesting an eastward extension of MJO propagation. We thank you for addressing that we may have overlooked other features of the MJO, and in response we have included an analysis of the impacts of a projected decrease in the zonal wavenumber of the MJO, informed by Rushley et al. (2019). Another change in the MJO which we are not looking at is a change to the meridional scale of the MJO, including any change in northward propagation. There is little previous work on this particular feature to inform perturbations.

Citations

Rushley, S. S., Kim, D., & Adames, Á. F. (2019). Changes in the MJO under Greenhouse Gas–Induced Warming in CMIP5 Models. *Journal of Climate*, *32*(3), 803–821.

3. The authors force the LBM using only the January condition. The authors may need to provide a strong reason why they are not using the entire winter season mean.

It is common in MJO teleconnection studies to use the three-month mean of December-Februrary (DJF) to represent boreal winter. Here, we choose instead to use Januaries for two main reasons:

A. There is some variation in the mean state winds between each of the boreal winter months. In the real world, propagating Rossby waves (which have a time scale of about 1-2 weeks) do not "feel" the seasonal mean winds, but rather winds that evolve on Rossby wave time scales.
B. We wanted to be able to adequately sample variations in the MJO teleconnection due to internal variability, as a way to then isolate the *consistent* change in the teleconnection. Using 30 DJFs instead of 30 Januaries would have reduced the variability of the winds used as input.

4. The authors only use one ensemble member from the CMIP6. However, there can be very large uncertainty coming from different ensemble members. Also, these ensembles may not be the ensembles that produce the most realistic MJO teleconnections in their historical runs. Is there a reason why the authors are not using the ensemble mean?

We use a single ensemble member instead of the ensemble mean from each model in order to ease the burden of data downloading on the author's part, and to keep the number of ensemble members used for each model consistent. Because we are comparing models, using the ensemble mean can confuse results when models have different numbers of ensemble members. For example, the ensemble mean from a model with a large number of ensemble members is less likely to be influenced by internal variability than a model with a fewer number of ensemble members. However, we acknowledge that having ensemble spread is desirable in order to separate the forced response from internal variability. It is for this reason that we run ensembles of 30 members (using a different year for each ensemble member) for each experiment and for each CMIP6 model's basic state. As an aside, internal variability may also contribute to model spread

We have added the following text to the methods,

"We use interannual ensembles in this way, rather than using the other ensemble members from each CMIP6 model, to minimize the data downloading burden and to keep the number of ensemble members used for each CMIP6 model mean state consistent"

5. When deciding the lower and upper bounds of the MJO changes, the authors used the estimate from CMIP5 models from previous studies. It makes the analysis inconsistent in this study. It would be better to estimate the MJO changes based on the CMIP6 models. CMIP6 produces a much more realistic MJO propagation (Ahn et al. 2020), the authors would need to discuss if there are any sensitivity of the results. Also, the authors did not show evidence why they choose to extend the MJO to 20° eastward.

Yes, it would have been much more desirable to use projected changes to the MJO from CMIP6 rather than CMIP5. However, to the authors' knowledge, detailed analyses of simulated end-of-century changes to the MJO in CMIP6 have not yet been published, whereas values for CMIP5 are available. Nonetheless, Zhou et al. (2020) indeed find increases in the MJO's eastward extent and heating intensity (although they do not publish exact quantities), which is consistent with our choice of perturbations. According to Figure 3 from Zhou et al. (2020), the eastward extension appears to be about 10 degrees (combination of CMIP5 & CMIP6) and between 10-50 degrees based on Figure 1a from Maloney et al., (2019). So, we chose an intermediate number of 20. However, we note that despite improvement in the simulation of the MJO in CMIP6, many models underestimate the metric used in Ahn et al. (2020) to quantify propagation over the maritime continent, and the popular eastward/westward power ratio used to assess MJO propagation in models [link]. As reviewer #3 put it, "Our current understanding of how the MJO itself is projected to change is incomplete," and thus these estimates are first guesses.

In response to this comment we have added the following to the manuscript,

*Regarding the choice of 20 degrees:*

"This value is roughly informed from results in \citet{Maloney2019} (CMIP5) and in \citet{Zhou2020} (CMIP5 and CMIP6)."

*Regarding the inconsistency between mean state from CMIP6 and MJO changes from CMIP5:*

1. "We use estimates of changes to the MJO from CMIP5, rather than CMIP6, due to the availability of published, detailed analyses of MJO changes for CMIP5. However, we note that despite recent improvement in the simulation of the MJO in CMIP6 \citep{Ahn2020} much work is needed to deepen understanding of future changes to the MJO, and thus these estimates are crude, first guesses."
2. "Secondly, our results are sensitive to the ideal forcing used to explore how a changing MJO may affect teleconnection changes. We have used a very idealized heating to represent the MJO which is constant across LBM simulations. Thus, despite using a range of perturbations of this heating to sample the range of possible changes to the MJO and subsequent impacts on teleconnections, we are lacking a comprehensive sampling of the inter-model variations in future MJO characteristics. Perturbations rely on projected changes from CMIP5, rather than CMIP6 (the latter of which has an improved MJO \citep{Ahn2020}) due to the availability of detailed published analyses of modeled changes to the MJO."

Citations

Ahn, M., Kim, D., Kang, D., Lee, J., Sperber, K. R., Gleckler, P. J., et al. (2020). MJO propagation across the maritime continent: Are CMIP6 models better than CMIP5 models? *Geophysical Research Letters*, *47*(11). https://doi.org/10.1029/2020gl087250

Maloney, E. D., Adames, Á. F., & Bui, H. X. (2019). Madden–Julian oscillation changes under anthropogenic warming. *Nature Climate Change*, *9*(1), 26–33.

Zhou, W., Yang, D., Xie, S.-P., & Ma, J. (2020). Amplified Madden–Julian oscillation impacts in the Pacific–North America region. *Nature Climate Change*. https://doi.org/10.1038/s41558-020-0814-0

6. The amplitude metric in this study may not be clean enough to represent the amplitude only. The metric can also be influenced by changes in the location of MJO teleconnections. Is there a reason why the authors not just simply calculate the spatial variance of MJO teleconnections as in Wang et al. (2020)?

We wanted to choose a metric that permitted the quantification of the evolving MJO teleconnection signal as the MJO propagated. Indeed, the metric may be influenced by changes to the location of MJO teleconnections, which we mention in the conclusions as a possible result that would need to be further investigated to examine more localized manifestations of the near-zero mean changes over the larger regions we have chosen. We believe that our metric, when viewed in map form as in Figure 2b-e, is appropriate for looking at changes to teleconnections (including the amplitude and pattern). However, when choosing the large North Pacific and North America regions to investigate, we did so intentionally so that we would *not* be focusing on changes to the pattern, and rather wanted to discuss more holistically changes across the region as a whole. Local changes to the pattern, such as in Zhou et al. (2020) are outside the scope of the present work.

Citations

Zhou, W., Yang, D., Xie, S.-P., & Ma, J. (2020). Amplified Madden–Julian oscillation impacts in the Pacific–North America region. *Nature Climate Change*. https://doi.org/10.1038/s41558-020-0814-0

7. The authors investigated the 850hPa meridional wind because they indicate that low-level circulation will be more important to regulate near-surface weather. However, the main purpose of this study is not to examine the MJO impacts on near-surface weather. Then it would be better to use either middle or upper tropospheric wind given that MJO teleconnections have larger amplitude over the middle to the upper troposphere and the results could be used to compare with previous studies.

We have re-written the paper using geopotential height at 500 hPa.

8. L232-234. Are these supported by CMIP6 projections? Here, it would be nice if the authors can compare their results with the CMIP6 projections.

Here, we are comparing the change in the teleconnection *due to changes in static stability alone*, which is not possible to assess using CMIP6 projections of teleconnection changes, which are also sensitive to the MJO and mean state wind. We have rephrased this sentence for additional clarity, below (bold = new).

"There is relatively good agreement in the teleconnection amplitude change **due to the change in the atmosphere's thermal structure alone (i.e., neglecting mean state wind or MJO changes)** between model mean states, particularly over the North Pacific."

9. Teleconnection changes are analyzed at the lower troposphere, whereas mechanisms (RWS and Ks) are analyzed at the upper troposphere. This can lead to inconsistency of the results. The authors may want to confirm their results by looking at teleconnection changes in the upper troposphere.

We have re-written the paper using geopotential height at 500 hPa.

10. The authors analyze the amplitude changes by separating the North Pacific and North America regions. Please provide a reason for looking into different regions.

We separated the Pacific-North America region into two smaller and separate regions because we did not want to assume that the amplitude changes in the two regions would be the same, so we analyzed them separately.

We have added the following,

"Here and throughout the remainder of this work, we choose to separate the larger Pacific-North America region into two smaller regions because we do not want to assume that the amplitude changes in the two regions are the same. "

11. Eq (3): Here, the stationary wavenumber is for the barotropic model and mean wind with no zonal variation and no meridional wind. That being said, this equation is not fully consistent with the basic state they used for the calculation. If the authors are using the full basic state, they would need to use the more complete format of stationary wavenumber (Li et al. 2015). The authors may need to justify why they neglect the zonal asymmetry and meridional wind in Eq (3).

While equation (3) does neglect meridional wind, it does not neglect zonal asymmetry. The zonal wind used in the equation is the full zonal wind, including its zonal asymmetries. In using equation (3), we are following a precedent set by many previous studies investigating the impact of the mean state wind on MJO teleconnections. These studies use the stationary wavenumber in the form that we have used, including Zheng and Chang (2020), Henderson et al., (2017), Wang et al., (2020), Tseng et al. (2020).

We have clarified in the text that the zonal wind used in equation 3 is the **temporal mean, full (including symmetric and asymmetric components) zonal wind**. Additionally, we have added the following sentence,

"Multiple previous studies have used the stationary Rossby wave number in this form (i.e., neglecting meridional wind) when exploring the impact of the mean state on MJO teleconnections (Henderson et al., 2017; Tseng et al.,2020b; Wang et al., 2020; Zheng and Chang, 2020)."

Citations

Henderson, S. A., Maloney, E. D., & Son, S. W. (2017). Madden-Julian oscillation Pacific teleconnections: The impact of the basic state and MJO representation in general circulation models. *Journal of Climate*, *30*(12), 4567–4587.

Tseng, K.-C., Maloney, E., & Barnes, E. A. (2020). The Consistency of MJO Teleconnection Patterns on Interannual Time Scales. *Journal of Climate*, *33*(9), 3471–3486.

Wang, J., Kim, H., Kim, D., Henderson, S. A., Stan, C., & Maloney, E. D. (2020). MJO Teleconnections over the PNA Region in Climate Models. Part II: Impacts of the MJO and Basic State. *Journal of Climate*, *33*(12), 5081–5101.

Zheng, C., & Chang, E. K.-M. (2020). The Role of Extratropical Background Flow in Modulating the MJO Extratropical Response. *Journal of Climate*, *33*(11), 4513–4536.

12. Fig. 3. The larger spread of MJO teleconnection amplitude could be due to the larger spread of mean state wind projection by CMIP6s. The authors may need to show 1) whether the wind projection is more uncertain in CMIP6 models than DSE. If this is the case, it is very easy to interpret Fig. 3 that the larger uncertainty in teleconnection amplitude is due to larger uncertainty in future mean wind projection, which is important to look at. 2) However if the projection uncertainty is similar between wind and DSE in CMIP6s, the authors may need to discuss why changes in the wind would lead to larger uncertainty in teleconnection amplitude than changes in DSE.

We have included a figure emphasizing the robustness of future increases to the dry static stability to hopefully increase the clarity that the large spread in the teleconnection change due to changes in the wind is a direct result of large spread in future projections to the winds.

Some figures that need to be improved or questions regarding the figures:

1. It would be quiet useful if the authors could provide the spatial pattern of MJO corresponding to their perturbed cases. Only the upper bound case would be sufficient to visualize the MJO pattern to facilitate understanding.

We have included a figure showing the perturbed MJO cases in the appendix.

2. Fig. 2a. Is this location arbitrary? Why are the authors not showing the multi-model mean of the CMIP6 which may be more useful?

The purpose of this panel is to illustrate how we are calculating the teleconnection amplitude metric. The location is arbitrary and is selected simply just to show how the field varies with time for each ensemble member using the ACCESS-ESM1-5 basic state in the LBM simulations. It would be inappropriate to use the results from each CMIP6 run because it would no longer be an accurate representation of how we calculate the teleconnection amplitude for a single model mean state. Not to mention, due to large intermodel differences in the mean state winds, the phase of the wave being depicted in Figure 2a would no longer be consistent between lines (which would now be individual model ensemble means). We have added the following to the manuscript, in reference to this panel (bold = new),

"Reassuringly, the propagating MJO-like forcing in the model excites a plausible time-varying response in the extratropics. **As an example**, Fig. \ref{fig:lbm_example}a shows the time-varying response of 500 hPa geopotential height at 60$^\circ$ N, 115$^\circ$ W for each ensemble member (thin light blue lines) and their mean (thick black line) for the set of 30 simulations performed with the full mean state given by historical values from a single representative CMIP6 model (ACCESS-ESM1-5). The smallness of the geopotential height anomalies relative to a typical observed midlatitude value is an expected consequence of the smallness of the amplitude used to force the model, which we kept small to try to minimize the number of unstable simulations.

We next introduce a scalar metric of MJO teleconnection strength that is appropriate to visualize in map form, recognizing that the magnitude of peaks and troughs of the ensemble mean response to the propagating forcing (**for example, the thick black line in Fig. \ref{fig:lbm_example}a**) is one way to quantify the magnitude of the consistent (ensemble mean signal that stands apart from the noise due to internal variability) teleconnection strength."

3. Please be very clear here that Fig. 2b is based on LBM response not based on real multi-model mean CMIP6.

We have edited the sentences introducing Fig 2b-e so that it is more clear that this figure uses output from the LBM.

4. Fig. 2b. very hard to see the strengthening (highlight of the contour). The authors did not provide the longitude of this eastern boundary to quantify the eastward extension. Also, it may be useful if the authors could provide the difference map with the control runs to help interpret the changes.

This strengthening is no longer apparent upon the switch to using geopotential height at 500 hPa. We have edited this Figure so the subsequent three map panels are differences from the first.

5. Fig. 3.

a. 1) No changes in DSE is shown, then how the authors know there is an increase in dry static stability? Is it based on previous studies? Some CMIP6 may project different changes in DSE. The authors may need to show how each CMIP6 model project DSE change and how that relates to teleconnection change. That being said, Fig.3 can be separated into 8 figures. In one figure, one axis represents DSE change in the CMIP6 models, one represents teleconnection amplitude change (either absolute change or fractional area). The same to the wind. By doing this, the authors could see how DSE or wind influences teleconnection amplitude.

The increase of the tropical dry static stability with surface warming is a robust thermodynamic response of the climate system to increased greenhouses gases. It is a result of the tropical lapse rate roughly following the moist adiabatic lapse rate, and the moist adiabatic lapse rate becoming more stable at higher temperatures. We mention this in the introduction, ("It is expected that the tropical static stability will increase in the future as the tropical temperature profile adjusts towards the moist adiabatic lapse rate of a warmer surface.") but in response to this comment, we have added another sentence in the results section where we discuss Figure 3:

"The increase in the tropical dry static stability is a consequence of amplified upper-tropospheric warming that occurs as tropical temperatures adjust to warmer surface temperatures \citep[e.g.,][]{Santer2005}."

We have also added a figure showing CMIP6 changes to the dry static stability.

b. 2) Also, the authors have never shown how DSE and wind are changing in CMIP6.

We have included a figure showing the change to the dry static stability.

c. 3) Very hard to separate the models with these markers here. Using numbers may be better to separate.

We tried recreating this figure with numbers instead of colored markers and found it more challenging to identify the models. We will keep the colored markers because we prefer the aesthetic but thank you for the suggestion. Additionally, we do not think the identification of the models is key for the interpretation of the paper.

[Figure]

6. Fig. 5b and c. Still not clear how these maps are calculated? What region is used for the average of the 15°x15° box? Also, the authors may want to only focus on the regions that have a high correlation coefficient. That means the authors can draw plots only with significant correlation in Fig. 5.

Thank you for the good suggestion. Upon the switch to geopotential height at 500 hPa, we no longer found a strong correlation between the local change in either Ks or the zonal wind and the regional (over the entire North Pacific or North America region) change in the teleconnection amplitude, anywhere. Nonetheless, we have updated the discussion of how these figures are created hopefully for increased clarity.

7. Fig. 6. Very hard to imagine why the spread can be this large over the North America region for Fig. 6e. The MJO is changing with the same propagation speed, and the mean state is not changing between the two runs. It may be helpful if the authors could provide the spatial maps of the two runs (historical MJO value and future MJO value) for these 10 models.

We have included a figure to the appendix showing how the mean state winds change for each individual model. We agree that the spread is indeed large, but the inter-model variations in the mean state winds are also quite large.

Some mechanisms that need more discussion:

1. L300-305. Better to show if this can be found in any CMIP6 model. Also, it is not clear why an increase in the number of Rossby waves can lead to stronger MJO teleconnections?

By "increase in the number of Rossby waves" we mean those excited by the MJO. Previous work (Tseng et al., 2020; Zheng et al. 2020) has shown that the jet structure in the east Pacific is very important for Rossby wave propagation through this region and to North America. When the subtropical jet shifts meridionally or in zonal extent, Rossby waves can be more/less likely to travel northward over the Gulf of

Alaska and across North America. This exact mechanism has been shown by Tseng et al. (2020) and Zheng and Chang (2020) to affect teleconnection strength over North America.

Nonetheless, this link disappeared when switching to geopotential height at 500 hPa (we also confirmed this with meridional wind at 300 hPa).

Citations

Tseng, K.-C., Maloney, E., & Barnes, E. A. (2020). The Consistency of MJO Teleconnection Patterns on Interannual Time Scales. *Journal of Climate*, *33*(9), 3471−3486.

Zheng, C., & Chang, E. K.-M. (2020). The Role of Extratropical Background Flow in Modulating the MJO Extratropical Response. *Journal of Climate*, *33*(11), 4513−4536.

2. L329-331. What is the underlying mechanism? The more eastward extent leading to larger teleconnection amplitude has been found in Adames and Wallace (2014).

Regarding the mechanism, we have not directly analyzed this. However, it is likely related to an eastward broadening of the region over which the propagating MJO can excite Rossby waves. Inter-model differences in the response to the increase in the eastward extent is directly related to the mean state. We think it is differences in the mean state winds driving inter-model differences (i.e., Rossby wave excitation and/or propagation), although this remains to be determined. We have added the following to the text (bold=new),

"Increasing the eastward extent of the propagating MJO thermal forcing in the LBM simulations increases the teleconnection amplitude for all mean states used and has the potential to produce the largest increases in the MJO teleconnection amplitude, more so over the North America region. **This is likely related to an eastward broadening of the region over which the MJO excites Rossby waves. \citet{Adames2014} found that for the mean state winds of the period from 1979-2011, the MJO excites the strongest extratropical response when heating maxima are located over the central Pacific. Inter-model differences between responses to an increase in the eastward extent are a direct consequence of differences in the mean state.**"

Thank you for the citation.

Minor comments:

1. The title is too misleading. Firstly, this paper discusses mainly the amplitude or strength of the MJO teleconnections which should be pointed out in the title. Secondly, the paper runs the simple model using the CMIP6 mean state but the results are not based on the CMIP6 projections, and the MJO heating is even idealized. So the authors may need to change the title to the numerical study of the mechanisms driving MJO teleconnection amplitude changes, etc. Also, this paper is more focused on the important factors contributing to the teleconnection amplitude changes rather than the mechanisms. Most mechanisms discussed in this study are based on findings in previous studies.

We have updated the title to, "Drivers of uncertainty in future projections of MJO teleconnections"

2. L5-6. For example? The more eastward extension seems to be a robust change in most models (Zhou et al. 2020).

Indeed! In writing this sentence, we were thinking more broadly about changes to teleconnections over the broader Pacific-North America region. The results from Zhou et al. (2020) are very limited in space (California only). We have updated the sentence as follows,

"Current state-of-the-art climate models do not agree on how MJO teleconnections over much of North America will change in a future climate."

3. This study is focused on which season? How MJO teleconnections are characterized? These would need to be mentioned in the abstract.

We have included this information in the abstract.

4. L9-10. This is found in Wolding et al. (2016), which would need to be mentioned here.'

Wolding et al. (2016) is a single-model study. This work is the first multi-model study confirming this result. We have updated the sentence as follows,

"We find that a weakening teleconnection due to increases in tropical dry static stability alone are robust across CMIP6 models"

5. L11-12. Also found in Wang et al. (2020), which would need to be mentioned here.

Wang et al. (2020) analyzes the current climate (1975-2005) in CMIP5 models. Here, when we mention "changes" we are referring to climate change. We have updated the sentence as follows (bold=new),

"We find no systematic relationship between **future** changes in Rossby wave excitation and the MJO teleconnection."

6. L12-13. Briefly explain more here about the mechanism.

This mechanism went away upon switching from v850 to z500.

7. L14-16. May be better to put it right after L11.

Done.

8. L16-17. This may not be a good argument. The reduction is implied by the LBM experiment forced by the CMIP6 mean state rather than in the majority of CMIP6 models.

We have updated the sentence as follows,

*"LBM simulations suggest a reduction of the boreal winter MJO teleconnection over the North Pacific, and an uncertain change over North America, with large spread over both regions that lends to weak confidence in the overall outlook. "*

9. L23-25. It is not clear how MJO teleconnections would be impacted by these factors. Since these are largely discussed in this study, a much more detailed introduction is needed to emphasize the motivation.

Given that we go into much more detail in the subsequent paragraphs, we have removed these sentences from this particular paragraph.

10. L26. How do the authors come to this conclusion? No references are discussed and cited here.

We have removed this sentence.

11. L26-27. Why only these two factors are mentioned among very many factors that influencing the MJO simulations? The authors may want to point out that MJO simulation is relatively poor in some CMIP5 models as that is the main focus in Ahn et al. (2017).

We have removed this sentence.

12. L28-29. Zhou et al. (2020) already provided some very good analysis of the teleconnection changes. The authors may want to discuss a bit about their results and what remains unknown. This sentence is not their main conclusion. Also, what is the relationship between this and the previous sentence?

We have removed the sentence before this one and have updated this sentence as follows,

*"While previous work suggests that over specific regions (such as the North American west coast) models agree on how MJO teleconnections will change in a future climate, over much of the North Pacific and over North America, it is unclear how the influence of the MJO will evolve with increasing atmospheric greenhouse gases \citep{Zhou2020}. "*

13. L30. "MJO teleconnection strength" over which region?

Globally. We have updated the sentence.

14. L31. L35-36. Need references.

We have added references.

15. L36-37. Discuss how "MJO precipitation intensity or cloud optical properties" would change

We point the reviewer to the subsequent sentences, which discuss the MJO's diabatic heating and precipitation,

"However, there is a considerable amount of inter-model spread, much of which is tied to disagreement in future projections of the MJO's diabatic heating and precipitation \citep{Bui2018, Maloney2019, Bui2019a, Bui2019b}."

16. L39-41. Do the authors mean the large spread in circulation projection is because of the spread in the projection of precipitation?

For CMIP5 models, Bui and Maloney (2018) show that there is a very close correspondence between modeled changes in the MJO precipitation and circulation strength. We have included their Figure 2 here. We use the language that the inter-model spread in future projections of the MJO's circulation strength is "tied to disagreement in future projections of the MJO's diabatic heating and precipitation" to carefully avoid using causal language. That is, the two are very closely associated/correlated, but this may not necessarily imply causality.

[Figure]

**Figure 2.** Differences in the 30- to 90-day standard deviation of precipitation (x-axis) and (a) 500 hPa omega and (b) 850 hPa zonal wind in RCP8.5 relative to the historical simulation. All the values have been normalized by the average between the historical and RCP8.5 simulations.

Citations

Bui, H. X., & Maloney, E. D. (2018). Changes in Madden-Julian Oscillation Precipitation and Wind Variance Under Global Warming. *Geophysical Research Letters*. Retrieved from https://agupubs.onlinelibrary.wiley.com/doi/abs/10.1029/2018GL078504

17. L41-43. what are the relationships between these two studies? One is talking about static stability, one is circulation change? Better to make consistency here although they may be coupled.

Yes the two are very tightly coupled. We have added a sentence to this paragraph to increase clarity about the two main controls on the circulation strength of the MJO: the dry static stability and the diabatic heating rate (bold = new).

"To first order, the rate that air rises and sinks in the MJO's circulation is tightly constrained by, and inversely proportional to, the tropical dry static stability \citep{Wolding2016}. Latent heat release associated with MJO precipitation is balanced by the upward advection of dry static energy. Similarly, radiative cooling in the MJO's dry region is balanced by slow, adiabatic subsidence. **That is, the MJO's circulation strength is directly proportional to its diabatic heating rate, and inversely proportional to the dry static stability.** "

18. L57-58. What changes in mean wind and what changes in teleconnections the authors believe would be important to look at? Is there any evidence from previous work which could serve as motivation?

We have modified the preceding sentences of this paragraph as follows,

"\citet{Tseng2020} and \citet{Zheng2020} show how variability in the mean state winds over the eastern North Pacific drives variability in the MJO teleconnection, with variations in the jet's zonal extension or meridional position playing an important role in modulating wave propagation, and hence MJO teleconnectivity, over North America. Finally, \citet{Zhou2020} show that modeled increases in the MJO's impact over the North American west coast with warming are due to an extension of the Pacific jet, which causes a shift in the MJO teleconnection pattern. However, little work has been done to explore the direct role that changes to the mean state winds have on changing MJO teleconnections to the broader North Pacific and North America region. "

19. L64-65. Is this found in CMIP6 projections? any reference? Or just the authors' hypothesis?

This has yet to be confirmed with CMIP6 results, and is our hypothesis based on previous work. We have modified the sentence as follows, (bold = new)

"**Comparisons between composites of observed geopotential height anomalies during fast and slow MJO events show differences in both the teleconnection pattern and magnitude of the anomalies \citep{Yadav2017}**...Many climate models predict an increase in the MJO's propagation speed with warming \citep[e.g.,][]{Rushley2019}, which, **considering previous work**, may contribute to a weakening, or at least a change in the pattern of the MJO teleconnection with warming."

20. L70-71. References or hypothesis? There are many other changes in MJO, such as frequency of events, the authors may want to point out why they are focused only on propagation speed and extent. Is it because these are the most robust changes? If so, please provide references.

This is a hypothesis. We have changed the words "will extend" to "may extend" to reflect this. We have added a paragraph to the introduction discussing other ways the MJO may change.

21. L73. Please clarify what the authors mean by "MJO teleconnection change". It has been quite confusing throughout the text. How the authors quantify the uncertainty? Please briefly specify here.

We have updated this sentence as, "While \citet{Zhou2020} show that the change in the MJO's impact on both the circulation and precipitation over most of North America is near-zero, they do not quantify the uncertainty in this projection over this larger region."

22. L79-81. It is better to have the main motivation to fill the gap of previous studies which authors should discuss more rather than from the prediction perspective which is too operational oriented.

We have updated the paragraph.

23. L81-83. This could sound like a big problem here. The authors may need to prove that this would not be an issue such as the authors are using an idealized MJO.

We have added the following sentence to the introduction in the subsequent paragraph,

"Finally, because we can separately prescribe an MJO forcing, it is not necessary to have a realistic internally generated MJO when investigating the sensitivity of the MJO teleconnection to inter-model variations in the mean state, thus permitting us to avoid only analyzing those CMIP6 models which produce more reliable MJOs."

24. L84. Please clearly describe here what analysis will use CMIP6 model output and what will use LBM. This is quite confusing in the introduction.

We point the reviewer to the methods section, which describes how we use CMIP6 model output with the LBM.

25. L85-86. Why the authors hypothesize the uncertainty in projections is large? I did not see this point in the previous paragraphs, only a bit was discussed in Zhou et al. (2020). But this cannot be considered as the authors' hypothesis. The authors may need to provide more evidence to support this hypothesis.

We have rephrased this sentence as, "In this study, we will use a linear baroclinic model \citep[LBM,][]{Watanabe2000} in concert with output from CMIP6 to separately quantify the contributions to uncertainty in future projections of MJO teleconnection by various mechanisms..."

26. L86. "various mechanisms" such as...? Please also briefly describe how to quantify the contribution?

We have updated the sentence as follows, "In this study, we will use a linear baroclinic model \citep[LBM,][]{Watanabe2000} in concert with output from CMIP6 to separately quantify the contributions to uncertainty in future projections of MJO teleconnection by various mechanisms: mean state changes and changes to the MJO's eastward propagation extent, propagation speed, heating intensity, and zonal wavenumber."

In reference to the comment about contributions, we point the reviewer to the methods and results sections where this is described in more detail.

27. Fig. 1. Any references that used similar idealized MJO heating?

Yes. We have added two references.

28. Please make the discussion consistent. Only use "amplitude" or only use "strength". Now is a bit confusing.

Where appropriate, we have changed the word "strength" to "amplitude" for consistency.

29. L217. Are the authors changing both the zonal and meridional mean winds in LBM?

We have included the following sentences in the methods for clarity

"The LBM takes as input three-dimensional winds, temperature, geopotential height, and surface pressure. The set of variables held constant when the "winds" are held constant are the zonal, meridional, and vertical wind. Similarly, the set of variables held constant when the "dry static energy" is held constant are the temperature, geopotential height, and surface pressure."

30. L222. How fractional area is calculated?

We have included the following sentence here,

"By fractional area, we mean the fraction of the region that shows a teleconnection amplitude change greater than zero."

31. L244. Can only say "changes in dry static stability". The authors did not show how DSE is changing in CMIP6s.

We have included a figure showing how the dry static stability (the vertical gradient of dry static energy) is changing.

32. The finding that RWS is not the key role in changing the teleconnection amplitude is also implied in Wang et al. (2020). They found that changes in the amplitude of RWS does not necessarily lead to the same changes in the amplitude of teleconnections.

We have included this reference in the discussion:

"This is consistent with \citet{Wang2020} who show that CMIP5 models with stronger $S'$ do not necessarily have stronger MJO teleconnections."

33. L290. What sizes of the boxes have the authors tried?

We have included this information in the text (3-20 degrees).

34. I am not sure why the experiments of the MJO intensity are necessary? It is very obvious that with a linear model the teleconnections would change as to how heating intensity changes.

Exactly, this is why we included the following sentences:

"The linearity of the LBM ensures that, in the absence of instability, the magnitude of the simulated response varies linearly with the magnitude of the forcing. We verified this with simulations using a 20\% increase in the magnitude of the forcing for two different model basic states. For the remainder of the 8 models used in these experiments, we thus did not run simulations (see Table \ref{table:exp}), because the magnitude of the extratropical response to the perturbed forcing amplitude is independent of the mean state. If nonlinear interactions had been considered, the variations across the mean states could have also been important in the response of the teleconnection to changes in heating intensity. "

We include this information in the figures for clarity & organization (i.e., to keep all of the information in one place). While it is not necessary to actually do the simulations, a change in the MJO's heating intensity is projected in simulations. Thus we include it here so that we can compare it to spread from the mean state.

    35. L315-316. Is this arbitrary? How the authors chose the models?

As we write in the next sentence,

"For these experiments, we chose CMIP6 models that produced a minimum number of unstable ensemble members in experiments used in Sect. 3.1 (see Fig. A1)."

    36. L326-329. Impacts of propagation speed on the teleconnection amplitude have been found in many previous studies already (e.g., Yadav and Straus 2017; Goss and Feldstein 2018; Zheng and Chang 2019; Wang et al. 2020).

Thank you for these references. We have added them to the discussion on previous work linking MJO propagation speed to MJO teleconnections in the introduction. We have modified the sentences referenced as follows,

"Consistent with previous work finding decreased MJO teleconnection strength with increased MJO propagation speed \citep{Zheng2019,Wang2020,Blade1995}, we find that increases to the propagation speed of our idealized heating produce modest decreases in the teleconnection amplitude over the North Pacific and North America."

    37. L333. "uncertainty in changes to the MJO". I do not understand why there is uncertainty in the MJO? Isn't the MJO difference the same between different models? The MJO used in the forcing is just an idealized MJO which is certain. "uncertainty in the mean state". Why there is uncertainty in the mean state? Isn't the mean state just used the historical value? Is here the uncertainty mainly comes from the model difference of their mean state?

The uncertainty in future projections of the MJO comes from previous work (i.e., analyses of changes in the MJO from CMIP5 used to inform the perturbations of the idealized forcing in our experiment). The uncertainty of the mean state refers to inter-model differences in future projections of the mean state: this discrepancy between models is a source of uncertainty. We have rewriten the paragraph to hopefully make this more clear.

"For perturbations to the MJO's propagation characteristics, Fig. \ref{fig:mjo_scatter} highlights the sensitivity of the MJO teleconnection change to the mean climate of each CMIP6 model. That is, for a given perturbation to the MJO's propagation speed, zonal wavenumber, or eastward extent, inter-model differences in the mean climate lead to variations in the teleconnection change. Thus, not only are changes in the mean climate important for understanding how the extratropical MJO teleconnection will evolve for an unchanging MJO, but also for understanding how MJO teleconnections will evolve in response to changes to the MJO. Additionally, for each individual MJO propagation or intensity characteristic, the spread in the MJO teleconnection change resulting from a perturbed MJO is on the order of the inter-model uncertainty that results from perturbations to the mean state alone (Fig. \ref{fig:bs_scatter}). These results challenge the notion that it may be possible to have currently have confidence in future projections of MJO teleconnections. "

38. L350-351. Be careful to make this conclusion as this is only a finding in the LBM.

We have rewritten the sentence as follows,

"These results suggest that over the North Pacific, mean state changes may lead to weaker MJO teleconnections."

References:

Lin, H., and G. Brunet, 2018: Extratropical response to the MJO: Nonlinearity and sensitivity to the initial state. J. Atmos. Sci., 75, 219–234.

Henderson, S.A. and Maloney, E.D., 2018. The impact of the Madden–Julian oscillation on high-latitude winter blocking during El Niño–Southern Oscillation events. Journal of Climate, 31(13), pp.5293-5318.

Ahn, M.S., Kim, D., Kang, D., Lee, J., Sperber, K.R., Gleckler, P.J., Jiang, X., Ham, Y.G. and Kim, H., 2020. MJO propagation across the Maritime Continent: Are CMIP6 models better than CMIP5 models?. Geophysical Research Letters, 47(11), p.e2020GL087250.

Wang, J., H. M. Kim, D. Kim, S. A. Henderson, C. Stan, and E. D. Maloney, 2020: MJO teleconnections over the PNA region in climate models. Part I: Performance- and process-based skill metrics. J. Climate, 33, 1051–1067.

Li, Y., Li, J., Jin, F.F. and Zhao, S., 2015. Interhemispheric propagation of stationary Rossby waves in a horizontally nonuniform background flow. Journal of the Atmospheric Sciences, 72(8), pp.3233-3256.

Adames, Á. F., and J. M. Wallace, 2014: Three-dimensional structure and evolution of the MJO and its relation to the mean flow. J. Atmos. Sci., 71, 2007–2026.

Yadav, P., and D. M. Straus, 2017: Circulation response to fast and slow MJO episodes. Mon. Wea. Rev., 145, 1577–1596.

Goss, M., and S. B. Feldstein, 2018: Testing the sensitivity of the extratropical response to the location, amplitude, and propagation speed of tropical convection. J. Atmos. Sci., 75, 639–655.

Zheng, C., and E. K. M. Chang, 2019: The role of MJO propagation, lifetime, and intensity on modulating the temporal evolution of the MJO extratropical response. J. Geophys. Res. Atmos., 124, 5352–5378

**Referee #3**

Review of " Mechanisms driving MJO teleconnection changes with warming in CMIP6"

By Jenney et al

The paper presents a diagnosis of changes in the strength of the Madden-Julian Oscillation (MJO) teleconnections in future climate simulations. The paper compares changes associated with an altered basic state (including static stability and the basic state winds) with those associated with a few selected characteristics of the MJO. The results also quantify the degree to which these factors differ in the various CMIP 6 models. The results are mostly convincing and merit publication. However the authors don't do a very good job of discussing the limitations and caveats of their study, but this should be relatively simple to fix.

General comments:

1. The authors mainly examine two aspects of how changes in the MJO itself might lead to an altered extratropical response, namely the propagation speed and eastward extent. Given the relatively poor simulation of the MJO in many previous models, it is not clear to me whether these are the only two possible changes that may occur. (I tend to have little faith in the future projections of the MJO in CMIP5 - maybe CMIP6 will be better – and most of the estimates of future behavior in the MJO seem to come from papers that analyze on the order of 10 CMIP5 models)

We have added a discussion of this limitation in the conclusions, which we describe further in reference to subsequent comments.

a. The authors use a very idealized heating perturbation in their equation 1, which clearly will not capture inter-model variability in how well the MJO is represented in the present climate, let alone the future climate. This caveat needs to be mentioned in the discussion.

We have added a paragraph that discusses the caveats of our idealized heating.

"Secondly, our results are sensitive to the ideal forcing used to explore how a changing MJO may affect teleconnection changes. We have used a very idealized heating to represent the MJO which is constant across LBM simulations. Thus, despite using a range of perturbations of this heating to sample the range of possible changes to the MJO and subsequent impacts on teleconnections, we are lacking a comprehensive sampling of the inter-model variations in future MJO characteristics. Perturbations rely on projected changes from CMIP5, rather than CMIP6 (the latter of which has an improved MJO \citep{Ahn2020}) due to the availability of detailed published analyses of modeled changes to the MJO. We also examine only a subset of the possible changes to the MJO that may occur; namely, its heating intensity, zonal wavenumber, propagation speed, and eastward propagation extent. For example, we have not investigated how changes to the MJO's frequency, meridional extent, or other changes to its spatial

structure may affect future MJO teleconnections. However, given that many models still struggle to reproduce a reliable MJO in the current climate, our current understanding of how the MJO may change with warming is incomplete. Thus there is much more work that needs to be done to understand how MJO teleconnections may change in the future."

b. The authors rightly note that their linear model is poorly suited for understanding whether a change in the magnitude of the MJO circulation might lead to a change in teleconnection, but it is conceivable that a stronger MJO circulation might lead to disproportionately stronger (or weaker) teleconnections if nonlinearities were allowed. While this is mentioned near line 320, this caveat needs to be mentioned more clearly both in the discussion and methods as well.

We have expanded this discussion in the conclusions as follows,

"A discussion of the caveats and limitations of our study is necessary. Firstly, we readily admit the limitations of our choice of a linear model to explore how the MJO teleconnection responds to changes in the mean state and to the MJO. While this allows us to easily diagnose causal relationships, span a previously sparsely sampled causative parameter space, and separately diagnose the effect of changes to mean state winds versus the dry static stability on MJO teleconnections using multiple model mean states; the linear framework is missing nonlinear relationships between the mean state, the MJO forcing, and MJO-excited Rossby waves. For example, the mean state exerts a strong control on MJO propagation characteristics \citep{Jiang2020}, which we have not included here. Rossby waves excited by the MJO are also able to extract energy from the mean flow \citep{Adames2014,Zheng2020}, and nonlinear interactions can cause a spatial shift in extratropical maximum geopotential height anomalies associated with the MJO \citep{Lin2018}. Both eddy-mean flow interactions, and any MJO dependence on the mean state are not simulated in the LBM. While previous work has shown that the MJO teleconnection is, to first order, linear \citep{Mori2008, Lin2018}, future work involving a nonlinear framework may be needed to reduce uncertainty in future projections of the MJO teleconnection to the extratropics. "

We have also added a discussion in the methods,

"Lastly, we emphasize that the linear framework we are using neglects nonlinear interactions in simulating MJO teleconnections. For example, in reality, the mean state exerts a strong control on MJO propagation characteristics \citep{Jiang2020}. Rossby waves excited by the MJO are able to extract energy from the mean flow \citep{Adames2014,Zheng2020}, and nonlinear interactions have been shown to lead to spatial shifts in extratropical teleconnection patterns \citep{Lin2018}. Thus, we caution that the results of this study are limited by the exclusion of these effects. "

c. The authors are relying on some previously published work on changes in the MJO, but other possible changes might occur in the MJO besides changes in amplitude, propagation speed, and eastward extent. If for example, the meridional extent of the MJO were to become broader as tropical GMS changes its structure under climate change, then the subtropical RWS could change its entire character. Whether this might occur is hard to know given the poor state of MJO's in CMIP models (at least CMIP5), but tropical GMS is indeed projected to change its structure and I tend to believe projections of, say, large scale GMS than of the MJO itself. The key point is that our current understanding of how the MJO itself is projected to change is incomplete.

We have added a paragraph to the conclusions that addresses this comment,

"Secondly, our results are sensitive to the ideal forcing used to explore how a changing MJO may affect teleconnection changes. We have used a very idealized heating to represent the MJO which is constant across LBM simulations. Thus, despite using a range of perturbations of this heating to sample the range of possible changes to the MJO and subsequent impacts on teleconnections, we are lacking a comprehensive sampling of the inter-model variations in future MJO characteristics. Perturbations rely on projected changes from CMIP5, rather than CMIP6 (the latter of which has an improved MJO \citep{Ahn2020}) due to the availability of detailed published analyses of modeled changes to the MJO. We also examine only a subset of the possible changes to the MJO that may occur; namely, its heating intensity, zonal wavenumber, propagation speed, and eastward propagation extent. For example, we have not investigated how changes to the MJO's frequency, meridional extent, or other changes to its spatial structure may affect future MJO teleconnections. However, given that many models still struggle to reproduce a reliable MJO in the current climate, our current understanding of how the MJO may change with warming is incomplete. Thus there is much  more work that needs to be done to understand how MJO teleconnections may change in the future. "

d. Another limitation is you only examine January mean state changes. The January Ks should feature more of a barrier in the West Pacific as compared to the shoulder months as the East Asian subtropical is strongest in January. It is conceivable that changes in the mean wind will matter less if you look at e.g. November or March. This limitation also isn't mentioned in the discussion.

We have included a short paragraph in the conclusions which addresses this. Thank you for the suggestion.

"Lastly, we have considered only January mean states for this work. It is possible that results could be different for other months or seasons. For example, changes in the mean wind may be less important during the shoulder months when the East Asian subtropical jet is weaker. "

More generally, the authors need to acknowledge in the discussion that they examine only a subset of the possible changes in the MJO that may actually occur. The current statement on lines 407-409 is very much overstated at present and needs to be deleted. I don't think the authors need to actually perform additional analysis to satisfy these point, rather add a paragraph or so that more fully discusses limitations.

We have included a paragraph in the conclusions which discusses that we examine only a subset of the possible changes to the MJO, and have deleted the referenced statement.

"Secondly, our results are sensitive to the ideal forcing used to explore how a changing MJO may affect teleconnection changes. We have used a very idealized heating to represent the MJO which is constant across LBM simulations. Thus, despite using a range of perturbations of this heating to sample the range of possible changes to the MJO and subsequent impacts on teleconnections, we are lacking a comprehensive sampling of the inter-model variations in future MJO characteristics. Perturbations rely on projected changes from CMIP5, rather than CMIP6 (the latter of which has an improved MJO \citep{Ahn2020}) due to the availability of detailed published analyses of modeled changes to the MJO. We also examine only a subset of the possible changes to the MJO that may occur; namely, its heating intensity, zonal wavenumber, propagation speed, and eastward propagation extent. For example, we have not investigated how changes to the MJO's frequency, meridional extent, or other changes to its spatial

structure may affect future MJO teleconnections. However, given that many models still struggle to reproduce a reliable MJO in the current climate, our current understanding of how the MJO may change with warming is incomplete. Thus there is much  more work that needs to be done to understand how MJO teleconnections may change in the future. "

2. Rossby wave source and Ks are both more clearly related to upper level metrics of the flow than lower level metrics. It is important to confirm that results are generally unchanged if some metric of the upper level flow is examined.

We have re-written the paper using geopotential height at 500 hPa instead of meridional wind at 850 hPa (and have separately confirmed the same results as z500 when we use meridional wind at 200 hPa). We find some minor changes, namely
-    Correlations between the change in the stationary Rossby wave number and the change in the teleconnection amplitude over North America interestingly disappeared.
-    More models showed increases in the teleconnection strength over North America. The final result is about the same, however (near-zero median change).

3. Figure 5a: This climatology of Ks doesn't look very much like that of Hoskins and Ambrizzi, their figure 3, especially in the subtropical East Pacific. If you plot Ks of reanalysis data, does the correspondence improve?

We have removed our plot showing the stationary Rossby wave number.

Nonetheless, our plot showing the multi-model January climatology of Ks for the historical period very closely resembles the December-February climatology from ERA-Interim published in Henderson et al. (2017). We include the two figures here for reference. The region of very small Ks in the subtropical eastern Pacific in reanalysis climatology does not appear in the figure we are showing in the manuscript because it is south of the latitude limits we have chosen for the figure.

[Figure]

[Figure]

[Figure]

FIG. 3. Stationary zonal wavenumber ($K_s$) derived from the 250-hPa Mercator zonal wind during DJF. Areas of easterly winds ($\bar{u}_M < 0$) are in white, and regions where $\beta_M < 0$ are in black.

Citations

Henderson, S. A., Maloney, E. D., & Son, S. W. (2017). Madden-Julian oscillation Pacific teleconnections: The impact of the basic state and MJO representation in general circulation models. *Journal of Climate*, *30*(12), 4567–4587.

Minor changes

Line 12 : I would replace "Rossy wave excitation" with "Rossby wave source", to be more precise

We have made the suggested change.

Line 87-88 this sentence is missing a few words

 We have rewritten the sentence.

Line 226 panel b hasn't been discussed yet

We have updated this sentence so we are no longer referencing the figure, and are now simply stating a general example as follows,

"For example, we find that there are some cases in which the mean teleconnection amplitude change over the region is positive, while the majority \textit{area} of the region shows a negative change."

Line 324 "models' mean-state" shouldn't this be amplitude of the MJO related heating?

Our intention here, with writing this particular sentence, was to acknowledge that, for a given specified increase in the MJO's heating intensity, variations in the mean state, when nonlinear interactions are permitted to occur, may lead to differences in the extratropical response between mean states. However, because we are using a linear model, variations in the mean state do not lead to variations in the response to a given perturbation of the MJO's heating intensity. It is perhaps inappropriate, as we had written, to say "if full nonlinearity had been considered", because this would in theory include interactions where mean state influences the heating intensity of the MJO. We have rewritten the sentence to avoid this confusion.

"If nonlinear interactions had been considered, variations across the different model climates could have also been important in the response of the teleconnection to changes in heating intensity. "

Line 326: "linearly sensitive to the mean state" – this was confusing to me

 We have deleted the word "linearly".

Line 322-323  "to changes to the MJO itself" I found these words confusing and unnecessary

We have removed all instances of the word "itself" where it appeared after the word "MJO".

---

## Referee Report (RR1)

Minor Revision:

(1) In the discussion of the experiments to test changes in the MJO characteristics, it is mentioned once only (line 167) that the MJO characteristic zonal wave number is decreased (zonal scale is increased). Thus an increase in the value of the x-coordinate in Figures 7(d)&7(f) (labeled "Wavenumber") correspond to a decrease in wave number. This is a bit confusing.The authors should emphasize the fact the wave number is decreased, and relabel the x-axis in panels (d) and (f) of Figure 7.
(2) The y-axis label to Figure 2(a) is wrong: it is geopotential height that is plotted, not meridional wind speed.

---

## Author Response (AR2)

Referee comments in black. Author responses to referee comments in red.
* * *
**Report #1 (Anonymous Referee #2)**

The manuscript is substantially improved compared to the last version. The discussion is more thorough and clear. The method is also clearly described and the results are convincing and interesting. I thus recommend acceptance for the publication after some minor revision.

Minor comments:
1. L6. "over much of North America". Better to provide the specific region.
We have made the suggested change.

2. L10-11. A little misleading. If I understand correctly, this conclusion is from the LBM simulations using the output from CMIP6? If this is the case, the authors may want to clearly point out the results are based on LBM forcing with the CMIP6 to avoid confusion.
We have made the suggested change.

3. It would be nice if the authors can discuss a little bit about the relative importance of the MJO and mean winds in the teleconnection uncertainty in the abstract. For example, which one is more important?
We have included the following sentence in the abstract,
"While quantitatively determining the relative importance of MJO versus mean state uncertainties in determining future teleconnections remains a challenge, the LBM simulations suggest that uncertainty in the mean state winds is a larger contributor to the uncertainty in future projections of the MJO teleconnection than the MJO."

4. L32-33. Better to clearly discuss how models agree on the future changes in MJO teleconnections over the NP. I see this is discussed in the later paragraphs, but it is better to also state here that the MJO teleconnections near the West Coast will be stronger? (Zhou et al. 2020)
We have made the suggested change.

5. Eq. (3). Please see Eq. (8) in Li et al. 2015 (reference provided in the first review comment). If zonal asymmetry is included in the zonal wind (as the authors argued here), zonal gradient of absolute vorticity will not be zero as in Eq. (3). This form already assumes that the zonal wind is zonally symmetric, although the authors used the full zonal wind to calculate Ks. The results are therefore inconsistent. If the authors can show that the zonal gradient of absolute vorticity is small might be helpful.
Many previous studies have used the same zonally asymmetric form of the stationary Rossby wave number that we have used here with the full (i.e., non-zonally uniform) zonal wind (see Equatin 2.4 in Hoskins and Ambrizzi, 1993). Eq. (3) does not assume that the zonal wind is zonally symmetric. As you note in the subsequent comment, we do ignore meridional wind. We have updated the following sentence (bold = new):
"Analyses of the stationary Rossby wave number have been useful for assessing how the mean state winds affect Rossby wave propagation
\citep[e.g.,][]{Henderson2017,Karoly1983,Tseng2020,Wang2020,Zheng2020}. The stationary Rossby wave number on a Mercator projection (to account for spherical geometry) **for a zonal flow** is defined as..."

Reference:
Hoskins, B. J., & Ambrizzi, T. (1993). Rossby Wave Propagation on a Realistic Longitudinally Varying Flow. *Journal of the Atmospheric Sciences*, 50(12), 1661–1671.

6. L387. Although previous studies generally used this form, the authors may still want to provide their own reason. Please see Li et al. 2015 who showed that meridional wind will be more important in determining the inter-hemispheric wave propagation. While here, the authors are more interested in the impacts of the zonal jet in NH only.

We have included the following to the section containing this content (bold = new):

"More work is needed to understand how changes to the mean flow will impact future MJO teleconnections, which our results indicate are likely a large source of inter-model spread in future projections of MJO teleconnections. **For example, consideration of meridional wind as in \citet{Li2015}, which is omitted in the form of the stationary Rossby wave number that we use, may reveal mechanisms due to changes to the meridional, rather than the zonal wind. We leave this for future work.**"

7. Section 4. The authors may want to point out their results are based on the LBM and may have difference if using CMIP6 simulations. A little more comparison with the results found in Zhou et al. 2015 will be helpful.

We have added the words "LBM-simulated" in multiple places in the conclusions to hopefully alleviate this confusion.

**Report #3 (Referee #1: David Straus)**

(1) In the discussion of the experiments to test changes in the MJO characteristics, it is mentioned once only (line 167) that the MJO characteristic zonal wave number is decreased (zonal scale is increased). Thus an increase in the value of the x-coordinate in Figures 7(d)&7(f) (labeled "Wavenumber") correspond to a decrease in wave number. This is a bit confusing.The authors should emphasize the fact the wave number is decreased, and relabel the x-axis in panels (d) and (f) of Figure 7.

We have added one more place where we clarify that the decrease in the zonal wavenumber corresponds to an increase in the zonal scale of the MJO. However, the horizontal axis that you point out in Figure 7 does not actually reference the change to the MJO, but rather the change in the teleconnection (as measured in geopotential height) that results from the prescribed change to the MJO. We point the referee to the following sentences where Figure 7 is introduced, which we have written to hopefully alleviate this confusion,

"Figure 7 summarizes the results of these simulations. As in Figure 4, the horizontal axis shows the regional mean difference in the teleconnection amplitude while the vertical axis shows the fractional area of the region that has stronger teleconnections (perturbed MJO compared to control MJO)."

(2) The y-axis label to Figure 2(a) is wrong: it is geopotential height that is plotted, not meridional wind speed.

Thank you for pointing out this error. We have fixed the label.